# Cortical reliability amid noise and chaos

Max Nolte [1], Michael W. Reimann [1], James G. King[1], Henry Markram[1,2] & Eilif B. Muller[1]

Typical responses of cortical neurons to identical sensory stimuli appear highly variable. It has thus been proposed that the cortex primarily uses a rate code. However, other studies have argued for spike-time coding under certain conditions. The potential role of spike-time coding is directly limited by the internally generated variability of cortical circuits, which remains largely unexplored. Here, we quantify this internally generated variability using a biophysical model of rat neocortical microcircuitry with biologically realistic noise sources. We find that stochastic neurotransmitter release is a critical component of internally generated variability, causing rapidly diverging, chaotic recurrent network dynamics. Surprisingly, the same non-linear recurrent network dynamics can transiently overcome the chaos in response to weak feed-forward thalamocortical inputs, and support reliable spike times with millisecond precision. Our model shows that the noisy and chaotic network dynamics of recurrent cortical microcircuitry are compatible with stimulus-evoked, millisecond spike-time reliability, resolving a long-standing debate.

[1] Blue Brain Project, École Polytechnique Fédérale de Lausanne, 1202 Geneva, Switzerland. [2] Laboratory of Neural Microcircuitry, Brain Mind Institute, École Polytechnique Fédérale de Lausanne, 1015 Lausanne, Switzerland. Correspondence and requests for materials should be addressed to M.N. (email: max.nolte@epfl.ch) or to E.B.M. (email: eilif.mueller@epfl.ch)

The typical electrical activity of cortical neurons is highly variable[1–4]. While part of this variability could be due to intrinsic noise sources, a substantial part could also be due to hidden variables such as unknown input from other parts of the brain, environmental parameters, or brain state[5–7]. Moreover, some neurons in sensory cortices can encode sensory input with high spike-time precision[8–10]. Taken together, it is compelling to assume that intrinsic noise plays a negligible role, and that cortical variability is essentially deterministic[11], encoding hidden or unobserved variables. This view is also supported by the fact that neocortical neurons respond to somatic current injections in vitro with high reliability[12]. However, there are two important reasons to believe that a large part of cortical variability is due to internally generated noise that carries no signal.

First, all cortical neurons are subject to cellular noise sources, such as stochastic synaptic transmission and ion-channel noise[13]. These noise sources originate from proteins susceptible to thermodynamic fluctuations and are truly intrinsic sources of noise[6,13]. In particular, synaptic transmission is based on a sequence of stochastic molecular events, where the low numbers of molecules involved do not allow stochastic properties to average out[14]. In tightly controlled slice conditions in vitro, the probability of vesicle release upon action potential arrival at a single cortical synapse is low (~50% between thick tufted layer five pyramidal neurons[15]), and estimated to be substantially lower in vivo[16] (~10% between same neurons[17]). The universal presence of synaptic noise suggests that cortical neurons respond far less reliably to presynaptic inputs than to current injections[18]. Furthermore, in vitro, some types of inhibitory neurons exhibit stochastic firing types[19]. That is, they respond highly irregularly to somatic current injections, due to amplified ion-channel noise[20].

Second, models suggest[21,22] and experiments show[23] that cortical networks have chaotic dynamics. This implies, by definition, that small perturbations, such as those due to intrinsic cellular noise, are amplified. Thus, extra or missing spikes in the network, for example, due to failed synaptic transmission, could fundamentally alter the trajectories of spiking activity in the network.

In spite of their potential importance, the separate and combined impacts of network dynamics and cellular noise sources on internally generated cortical neuronal variability remain largely unexplored, as it is currently impossible to measure all external inputs to a local population of cortical neurons in vivo. As a result, we are still unable to quantify how much of the experimentally observed variability is generated internally by the local circuitry, and how much is generated externally. Here, we addressed these questions with a recently developed simulation-based approach, namely a biologically constrained model of a prototypical neocortical microcircuit in rat somatosensory cortex (the NMC-model)[17]. The model consists of 31,346 neurons, ~8 million connections, and ~36 million synapses (Fig. 1a), and incorporates several prominent sources of noise such as stochastic synaptic transmission—including failure of vesicle release and spontaneous release—and ion-channel noise (Fig. 1b). Each of the noise sources is constrained to replicate experimentally observed variability. This bottom-up modeling approach provides control over all noise sources, as well as external inputs and internal states.

Through a series of simulation experiments, in which we selectively enable noise sources and recurrent network dynamics, we characterize internally generated cortical variability and how it arises. When cellular noise sources are disabled, we find that the underlying deterministic network dynamics are chaotic, whereas when noise sources are enabled, an interplay of stochastic synaptic transmission and network dynamics determines the rate by which membrane potentials diverge. Surprisingly, our model predicts that the recurrent cortical circuitry can transiently overcome these noisy and chaotic network dynamics in response to thalamocortical inputs and produce reliable patterns of activity.

## Results

**Rapid divergence of spontaneous activity.** Owing to the presence of intrinsic noise sources (Fig. 1b), neurons in the NMC-model are constantly perturbed. Combined with chaotic network dynamics, this could lead to highly variable activity trajectories. To assess the variability of activity trajectories, we quantified how spontaneous neuronal activity diverges from identical initial conditions. We simulated independent trials of network activity up to a time $t_0$, at which point we saved the full dynamical state of the simulation (*base state*). We then resumed the simulation two times from the base state, i.e., we used identical initial conditions and histories in each case, but with different sequences of random numbers. This allowed us to obtain two equally valid probabilistic network trajectories for $t > t_0$ for each base state. We observed that somatic membrane potentials ($V_m$) for individual neurons, and the mean potentials for the population both diverged rapidly between the two simulations (Fig. 1c).

To quantify the time-course of the divergence, we calculated the average root-mean-square deviation $RMSD_V(t)$ of somatic membrane potentials of individual neurons between two trials in time bins of size $\Delta t$ starting from $t_0$ (see Methods). We observed that $RMSD_V(t)$ diverged rapidly from zero and eventually converged towards a steady-state value $RMSD_\infty$, equal to the $RMSD_V$ of independent trials that did not share the same base state (Fig. 1d, solid black and dashed gray lines). The divergence was fast, with $RMSD_V(t)$ reaching > 50% of its steady-state value within 20 ms.

While the $RMSD_V(t)$ of the circuit allowed us to accurately track the overall divergence of the whole circuit, $RMSD_V(t)$ of individual neurons and trials was too noisy for in-depth analysis (Supplementary Fig. 1a1, b1). We note that while $RMSD_V(t)$ quantifies the absolute distance between membrane potentials, potentials can still be correlated independent of this distance. To this end, we analogously computed the average linear correlation $r_V(t)$ of somatic membrane potentials of individual neurons between two diverging trials. We found that the mean correlation $r_V(t)$ diverged faster than the absolute distance as measured by $RMSD_V(t)$ (Fig. 1d, blue line), again with a broad distribution across individual neurons (Supplementary Fig. 1a2, b2).

To better evaluate the difference between $r_V(t)$ and $RMSD_V(t)$, we defined the similarity $s_{RMSD}(t)$ of the microcircuit activity as the normalized difference between diverging and steady-state $RMSD_V(t)$ (and similarly $s_r(t)$ for $r_V(t)$). When similarity $s_{RMSD}(t) = 1$, membrane potential traces are identical; when $s_{RMSD}(t) = 0$ membrane potentials have reached their steady-state distance $RMSD_\infty$. Similarly, when $s_r(t) = 1$, membrane potentials have a perfect linear relationship; when $s_r(t) = 0$, they reached their steady-state correlation $r_\infty$. Comparing $s_r(t)$ and $s_{RMSD}(t)$, we observed that $r_V(t)$ diverged approximately twice as fast as $RMSD_V(t)$ (Fig. 1e1 vs. 1e2). More precisely, an exponential fit to the first 40 milliseconds revealed divergence time constants of $\tau_{RMSD} = 22.7 \pm 0.5$ ms and $\tau_r = 11.5 \pm 0.2$ ms ($\pm$ 68% confidence interval of fit). These were conserved for different bins sizes $\Delta t$ in the range of 1 ms to 50 ms (Supplementary Fig. 2b1, 2). However, a simple exponential decay does not provide an adequate description of the whole time-course of the similarity, as the time constant changes continuously, especially in the first several milliseconds (Supplementary Fig. 2a). A small but statistically significant difference ($p < 0.025$; one-sided $t$-test) between diverging and independent activity persisted for around 400 ms for

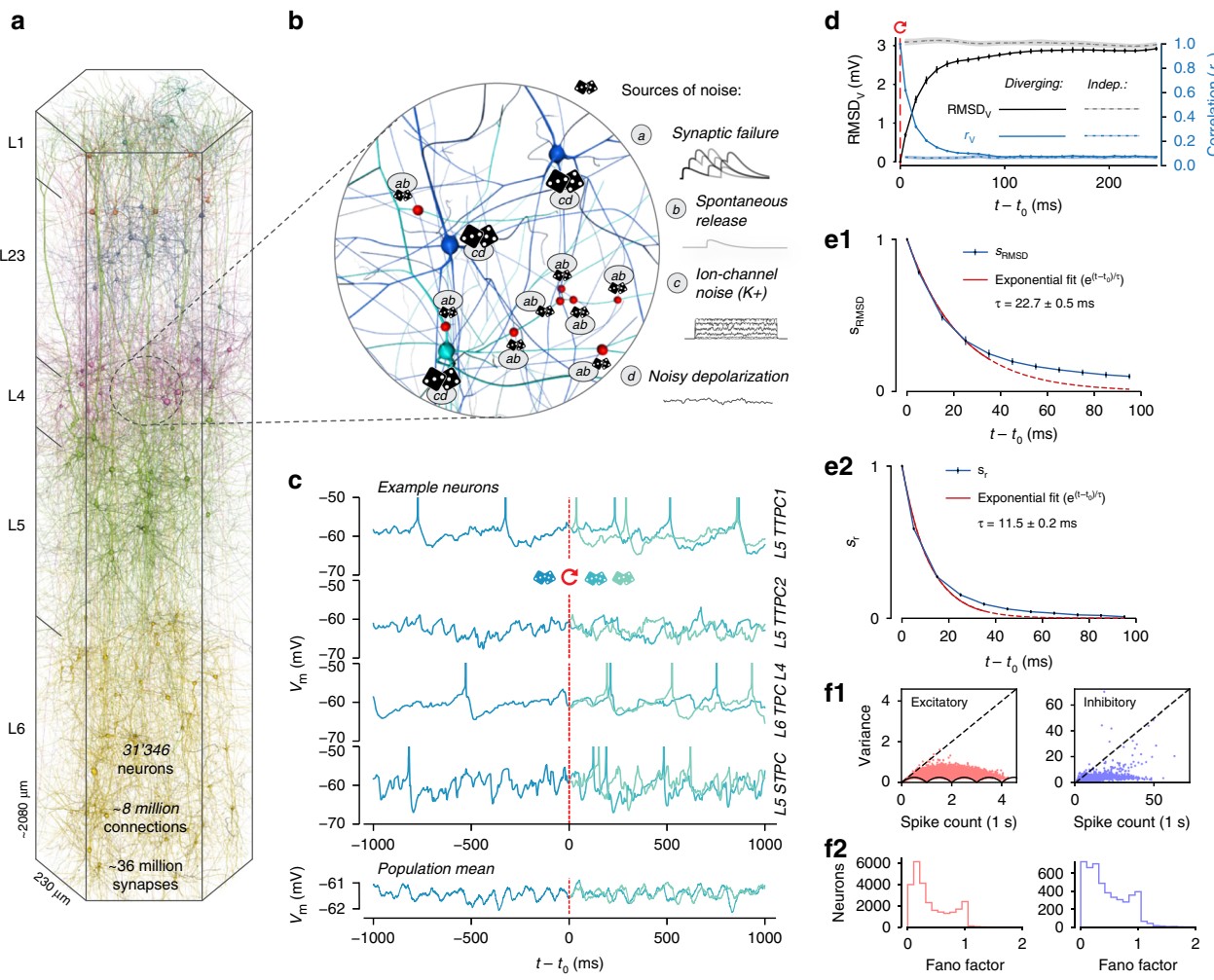

**Fig. 1** Rapid divergence of spontaneous activity. **a** Morphologically detailed model of a neocortical microcircuit (NMC); depicted are 100 randomly selected neurons, out of 31,346 in total (~0.3%). Neurons are colored according to their layer. **b** Examples of simulated noise sources in the NMC-model: stochastic synaptic transmission, including (*a*) vesicle release failure and (*b*) spontaneous vesicle release ("miniature PSPs") at all 36 million synapses; (*c*) probabilistic opening and closing of voltage-gated potassium channels in irregularly spiking inhibitory neurons (1137 out of 31,346 neurons); (*d*) a constant depolarizing current with a weak white noise component ($\sigma_s^2 \ll \mu_s$) injected into the somata of all neurons. **c** The membrane potential of four sample neurons (and population mean of all 31,346 neurons) during a network simulation of spontaneous activity. At $t_0$, the state of the microcircuit is saved, and then resumed twice with identical initial conditions, but with different random seeds for all noise sources. **d** Root-mean square deviation ($RMSD_V(t)$) and correlation ($r_V(t)$) of the somatic membrane potentials between pairs of resumed simulations diverging from identical initial conditions (mean of all neurons and 40 saved base states ± 95% confidence interval). The dashed lines depict the steady-state $RMSD_V$ and $r_V$ between independent simulations (i.e., resumed from different base states). **e** The similarity of the system ($s_{RMSD}$ and $s_r$) defined as the difference between the diverging and steady-state $RMSD_V$ and $r_V$, normalized to lie between 1 (identical) and 0 (fully diverged) (mean ± 95% confidence interval). Exponential fit of $s_{RMSD}$ and $s_r$ for $t - t_0 <$ 40 ms (estimated time constant ± 68% confidence interval of fit). **f1** Mean spike count and variance of spike count of 40 independent trials of 1000 ms duration for all neurons in the microcircuit, plotted separately for excitatory neurons (red) and inhibitory neurons (blue). The dashed lines indicate the expected values for a Poisson process. Black lines indicate minimum variance due to the fact that the spike count has to be an integer. **f2** Distribution of Fano factors (variance divided by mean spike count) corresponding to **f1**

$RMSD_V$ (Supplementary Fig. 2c1) and around 200 ms for $r_V$ (Supplementary Fig. 2c2).

We have shown that spontaneous activity in the NMC-model is highly variable, with rapidly diverging spontaneous activity trajectories both in terms of membrane potentials, and consequently spike times (see Fig. 1c and Supplementary Fig. 3a). The rapid timescale of divergence could imply that spike-count variability is high, akin to a Poisson process. However, the Fano factor (variance of spike counts divided by mean spike count) was far lower than for a Poisson process (Fano factor = 1)[18] for most neurons, especially for larger firing rates (Fig. 1f and Supplementary Fig. 4). Consequently, our model predicts that

Poisson-like spike-count variability is not generated internally within a microcircuit, and shows that rapidly diverging activity does not automatically lead to large spike-count fluctuations, likely as a result of spike frequency adaptation[24] and synaptic adaptation mechanisms[25].

**Robust divergence across dynamical states and circuit scale.** In addition to the microscopic divergence of individual somatic membrane voltages, macroscopic fluctuations in population spiking activity (Fig. 2a1), and population firing rate (Supplementary Fig. 3a) also diverged rapidly for $t > t_0$. The nature of these global fluctuations depends on the balance between

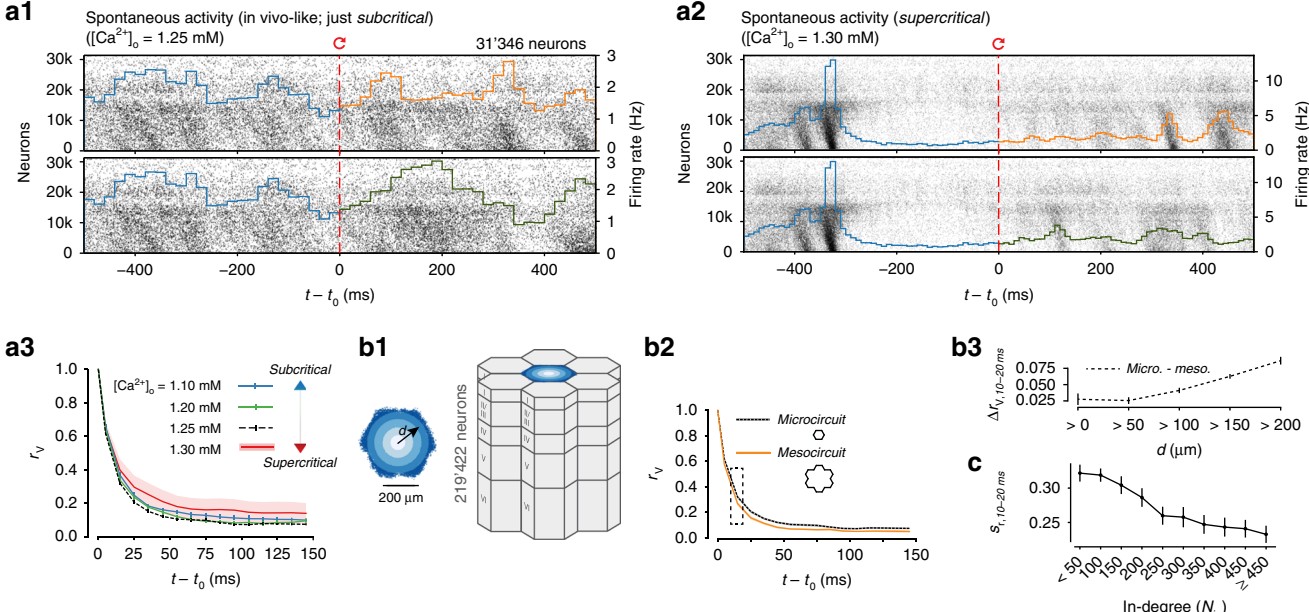

**Fig. 2** Robust divergence across dynamical states and circuit scale. **a1** Population raster plot and population peristimulus time histogram (PSTH) for all 31,346 neurons in the microcircuit, during spontaneous activity. Neurons are ordered according to cortical depth, with deep layers at the bottom and upper layers at the top. Each row represents the spikes of one neuron. For visibility, raster lines extend over dozens of rows for each neuron. For $t < t_0$, the top and bottom raster plots show the same simulation, whereas for $t > t_0$, the raster plots depict two simulations resuming from identical initial conditions at $t_0$, but using different random number seeds. **a2** Same as **a**, but for supercritical activity. **a3** $r_V$ across dynamical regimes (20 saved base states, mean ± 95% confidence interval; same as Fig. 1d for $[Ca^{2+}]_o = 1.25$ mM). **b1** The microcircuit (center, blue), surrounded by six other microcircuits (gray), forming a continuous mesocircuit of ~220,000 neurons, with no boundary effects between the circuits. **b2** $r_V$ for the center microcircuit when simulated without surrounding circuits (black), and of the center microcircuit when simulated as a mesocircuit (orange) (microcircuit: 40 saved base states; mesocircuit: 20 saved base states; mean ± 95% confidence interval). **b3** Quantifying edge effects. Difference of $r_V$ between the same neurons in the microcircuit and the mesocircuit at 10–20 ms, plotted according to distance from horizontal center (mean ± 95% confidence interval). **c** Similarity $s_r$ at 10–20 ms for subsets of neurons grouped by in-degree (bin size: 50; mean ± 95% confidence interval)

excitatory and inhibitory activity (EI-balance) in the network[26]. In the NMC-model, the EI-balance is determined by the integrated anatomical and physiological data, and can be modulated by changes in extracellular calcium concentration ($[Ca^{2+}]_o$) through its effect on synaptic vesicle release probabilities[16,17].

In the state analyzed here ($[Ca^{2+}]_o = 1.25$ mM), the microcircuit is in a just subcritical[27] state of asynchronous spontaneous activity, where it reproduces spontaneous and evoked network dynamics observed in vivo[17]. While this asynchronous state might be important for efficient coding[28,29], the exact EI-balance in vivo is difficult to determine, and is likely to reconfigure dynamically as a function of the state of arousal and attentiveness of the animal[30]. We therefore investigated the relationship between the time-course of divergence and different dynamical regimes, from subcritical asynchronous activity to supercritical synchronous activity (Fig. 2a2). We observed that the rapid divergence of electrical activity was approximately conserved across these different dynamical states (Fig. 2a3).

We further found that the divergence timescale was nearly saturated at the scale of the microcircuit, with only small changes compared to simulating a larger circuit (Fig. 2b1, 2). Only neurons at the periphery of the microcircuit diverged faster due to additional synaptic inputs (Fig. 2b3), as the number of synaptic inputs directly shapes the divergence (Fig. 2c) (see Supplementary Note 1, Supplementary Fig. 3b).

We note that $RMSD_V(t)$ and $r_V(t)$ are generally highly correlated (Supplementary Fig. 5a, *abcd*). In what follows, we hence present the divergence in terms of $r_V(t)$, except when there is a qualitative difference.

**Noise amplified by chaos determines divergence**. We have demonstrated a high level of variability, which is robust across dynamical states and nearly saturated at the scale of the microcircuit. Next, we studied how the interaction of cellular noise sources and recurrent network dynamics shapes this emergent variability. To this end, we performed two complementary sets of simulation experiments. In the first set, we sought insights into the role of network dynamics without noise sources, probing the sensitivity of a completely deterministic version of the model to a weak, momentary perturbation. In the second, we studied the opposite case of variability due to cellular noise sources without amplification by the network.

To implement the first set of simulations, we disabled stochasticity of cellular noise sources, including synaptic transmission, by using a fixed sequence of random numbers, which made the random outcome deterministic. This enabled us to observe amplification of perturbations through the network without the effect of continuously varying cellular noise sources. As the sole source of perturbation, we injected a single extra spike into one of the neurons in the microcircuit. We observed that the network diverged rapidly (Fig. 3a1, dashed line), though more slowly than with noise sources enabled (Fig. 3a1, solid line). In fact, even a miniscule current injection, which shifted the majority of spike times by < 0.05 ms, eventually led to a divergence of membrane potentials similar to the divergence observed in the full model with noise sources (Fig. 3a1, dotted line). The slightly higher steady-state correlation $r_\infty$ in the deterministic simulation was due to identical spontaneous release of neurotransmitter, identical ion-channel opening probabilities,

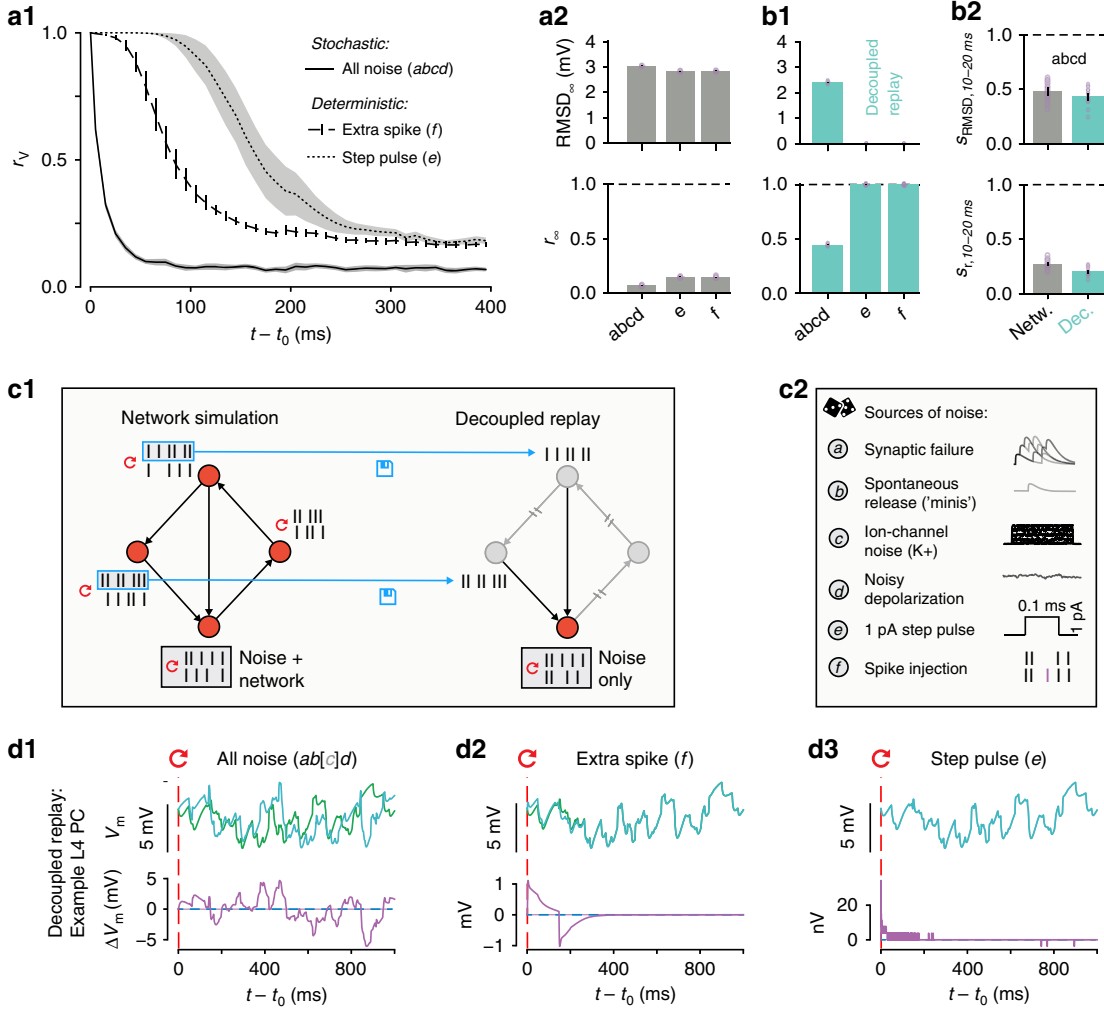

**Fig. 3 Noise amplified by chaos determines divergence. a1** Time-course of correlation $r_V$ after resuming at $t_0$ from identical conditions with different forms of perturbation. Full cellular noise as before, solid line (*abcd*); no cellular noise, but perturbing with a single extra spike in one neuron, dashed line (*f*); a miniscule step-pulse perturbation in all neurons, dotted line (*e*). (*abcd*: 40 saved base states; *e, f*: 20 saved base states; mean ± 95% confidence interval). **a2** Steady-state root-mean square deviation $RMSD_\infty$ and correlation $r_\infty$ for stochastic (*abcd*) and deterministic simulations (*e, f*) as defined in **a1** (mean ± 95% confidence interval in black; individual base states in purple dots). **b1** As in **a2**, but for decoupled, replayed simulations. (20 saved base states). **b2** Similarity $s_{RMSD}$ and $s_r$ at 10–20 ms with all noise sources enabled, for network and decoupled simulations (mean ± 95% confidence interval). **c1** Decoupled replay paradigm. Presynaptic spike trains from a network simulation are saved and then replayed to the synapses of each neuron in a decoupled simulation, thereby removing variability due to feedback network dynamics. **c2** Overview of sources of noise and perturbations. **d** Decoupled replay simulations (see **c1**) for a representative L4 PC neuron, with somatic membrane potential differences between the two trials only due to cellular noise sources (*ab[c]d*), a single extra presynaptic spike (*f*) or a miniscule step-pulse perturbation (*e*). [c] indicates that for some neuron types in the NMC-model, such as L4 PCs, no stochastic ion channels are present

and the small, but identical, noisy component of the depolarizing current injection. However, the relative difference in $RMSD_\infty$ was much smaller than the difference in $r_\infty$ between the deterministic and the stochastic simulations (Fig. 3a2, top vs. bottom). That is, any perturbation to the system eventually led to similarly large steady-state fluctuations. We conclude that the underlying dynamics of the circuit are chaotic, in the sense that small perturbations, such as one injected spike, lead to completely different activity trajectories.

It is important to note that when using a fixed random seed to make the stochastic version of the Tsodyks–Markram synapse model deterministic[17,31], any extra or missing presynaptic spike can change the outcome for the next spike by advancing the sequence of random numbers. To avoid this difficulty, we ran equivalent simulations using the deterministic version of the Tsodyks–Markram synapse model. In these simulations, extra

spikes and small perturbations produced qualitatively similar divergence time courses (Supplementary Fig. 6a vs. 6b, dark green and pink lines).

We have shown that the network amplifies extra spikes or even small perturbations of membrane potentials. This leads to chaotic divergence of activity with similar steady-state variability, but different time courses. It remained to be seen whether this high level of variability requires network amplification or whether it could be generated by the noise sources alone.

To address this question, we implemented a second set of simulations to study the case of ongoing noise sources without network propagation. In these *decoupled replay* simulations, in contrast to regular *network simulations*, synaptic mechanisms were activated by spikes at fixed times, recorded in an earlier simulation experiment (Fig. 3c1). In this way, the network was no longer able to amplify neuronal variability and neuronal

variability was entirely due either to cellular noise sources or perturbations (Fig. 3d). We found with all noise sources turned on, somatic membrane potentials still diverged rapidly, as quantified by the similarity $s_r$ at 10–20 ms (Fig. 3b2) (we found $s_{r,10\text{-}20\,ms}$ to be a good predictor of the relative order of $s_r$ at any time). However, steady-state $r_\infty$ was higher and $RMSD_\infty$ was lower than in the network simulations (Fig. 3b1 vs. 3a2). When the decoupled replay paradigm was used with the deterministic version of the model, single extra spikes and brief current injections only evoked small, transient perturbations (Fig. 3d2, 3). It follows that the high level of variability observed in network simulations was due to rapid perturbations of activity from cellular noise sources that were amplified by chaotic network dynamics.

**Synaptic noise dominates divergence**. To understand the contribution of individual noise sources in this interplay of noise and recurrent network dynamics, we designed a series of simulation experiments where we selectively disabled specific subsets of noise sources. We observed that disabling all noise sources except synaptic failure produced a time-course for $r_V(t)$ and steady-state divergence $r_\infty$ that was similar to observations with all noise sources combined (Fig. 4a1, black and green lines). On the other hand, disabling all but ion-channel noise or all but the noisy current injection led to much slower divergence (Fig. 4a1, orange and purple lines). As before, we quantified the speed of divergence by the similarity $s_r$ at 10–20 ms after $t_0$ ($s_{r,10\text{-}20ms}$) (Fig. 4a3, cyan). Our results suggest that simulations with synaptic failure give rise to rapid divergence, whereas steady-state $r_\infty$ and $RMSD_\infty$ depend on noise sources only weakly (Fig. 4a2). We conclude that in the NMC-model, the time-course of divergence depends on synaptic noise, a combination of synaptic failure and spontaneous release.

**Ion-channel noise is overshadowed by synaptic noise**. Synaptic noise in the NMC-model is modeled at every single synapse, while ion-channel noise is limited to irregular firing e-types[17,19]. Irregular e-types are defined by high-intrinsic spike-time variability in response to constant current injections in vitro, even in the absence of synaptic noise. In the NMC-model, irregular spiking is modeled with a subset of stochastic ion channels, in accordance with in vitro findings on the source of the irregular spiking patterns observed in cortical interneurons[20]. In contrast, regular firing e-types do not require noisy ion channels to replicate in vitro spiking behavior. To better understand the interplay of ion-channel noise and synaptic noise, we focused our next analysis solely on irregular firing e-types. We observed that irregular firing e-types diverged significantly faster than the whole population (Fig. 4a3, orange vs. cyan). However, synaptic noise still dominated over ion-channel noise. Enabling ion-channel noise in addition to synaptic noise led to only marginal gains in divergence rate; when ion-channel noise was enabled on its own, divergence was significantly slower (Fig. 4a3, orange, *ab* vs. *abcd* and *c*). This suggests that in in vivo conditions, noise from stochastic ion-channels is overshadowed by synaptic noise.

**Synaptic noise acts as a threshold for other noise sources**. Several smaller noise sources are not included in our model (see Discussion). To understand how additional noise sources of various magnitudes could influence divergence, we quantified the somatic voltage fluctuations due to the previously used combinations of cellular noise using the decoupled replay paradigm, i.e., with network propagation removed ($RMSD_{\infty,dec}$) (Fig. 4b; see Fig. 4c1–3 for representative examples). We found that the rate of divergence in a network simulation, $s_{RMSD,10\text{-}20ms}$, is inversely

proportional to $RMSD_{\infty,dec}$ (Fig. 4a3; see also Supplementary Fig. 7 for an extensive comparison of noise sources across simulation paradigms). In the NMC-model, synaptic noise leads to the largest $RMSD_{\infty,dec}$ and determines the rate of divergence.

How strong would missing cellular noise sources have to be to increase network variability beyond the level due to synaptic noise? To answer this question, we studied how the magnitude of a generic white noise depolarizing current (Fig. 4d1) affects the time-course of divergence. We found that this magnitude, $RMSD_{\infty,dec}^{d_x}$, needs to be larger than ~1 mV to impact divergence beyond synaptic noise (Fig. 4d2), and we therefore predict that synaptic noise is the most important cellular noise source (see Supplementary Note 2 and Supplementary Fig. 8).

**Rapid divergence of evoked reliable activity**. In the NMC-model, thalamic inputs can evoke responses with varying degrees of reliability[17,32]. What then are the roles of synaptic noise and chaotic network dynamics during these evoked responses? To answer this question, we simulated electrical activity in response to a naturalistic thalamocortical stimulus (Fig. 5a1), consisting of spike trains recorded in the ventral posteromedial nucleus (VPM) during replayed whisker deflection in vivo[33]. These spike trains were then applied to different feed-forward VPM fibers in the model to achieve a biologically inspired, time-varying synchronicity among inputs (Fig. 5a3). To avoid introducing external variability on top of the internally generated microcircuit variability, presynaptic inputs were kept identical across trials, but with thalamocortical synapses subject to the same synaptic noise as cortical synapses. Since this condition excludes variability in the system up to and including the thalamus, it can be considered an intermediate stage between the decoupled replay and regular network simulations. The simulations allowed us to identify an upper bound on the reliability of thalamocortical responses. Mean $r_V(t)$ during evoked activity was stronger than during spontaneous activity, moving between ~0.1 and ~0.4 (Fig. 5a2), indicating that external input increases neuronal reliability.

To characterize the nature of chaotic network dynamics during this evoked, reliable activity, we again resumed from identical initial conditions, with $t_0$ at various times relative to the stimulus onset at $t = 0$ ms (Fig. 5b, for $t_0 = 100$ ms). The population spiking activity across pairs of trials after resuming appeared almost identical, even for time intervals much larger than the divergence time characterized above (Fig. 5b). At first glance, it would appear that the input had fully overcome the chaotic divergence. However, quantification of variability by time-course of divergence of membrane potentials, $r_V(t)$, showed that it dropped rapidly towards the independent trial average (Fig. 5c1, top). When we resumed from identical initial conditions at different times, for example, at the onset of evoked activity (Fig. 5c1, middle) or before onset (Fig. 5c1, bottom), $r_V(t)$ dropped in the same way, subsequently converging to the average for independent trials. Indeed, $s_r(t)$, the normalized difference between the resumed and independent $r_V(t)$ showed a pattern of divergence remarkably similar to the divergence observed in simulations of spontaneous activity (Fig. 5c2). Resuming from a base state at the peak of evoked activity, $s_{RMSD}(t)$ drops even faster (Supplementary Fig. 9a). A simpler stimulus, designed to imitate a whisker flick-type experiment[17], yielded comparable results (Supplementary Fig. 9b, c). Hence, any neuronal activity, whether spontaneous and unpredictable, or evoked and reliable, is subject to similar divergent network dynamics.

**Evoked reliable activity amid noise and chaos**. At first glance, our observations of evoked reliable activity and chaotic divergence of membrane potentials seem to be contradictory. To better

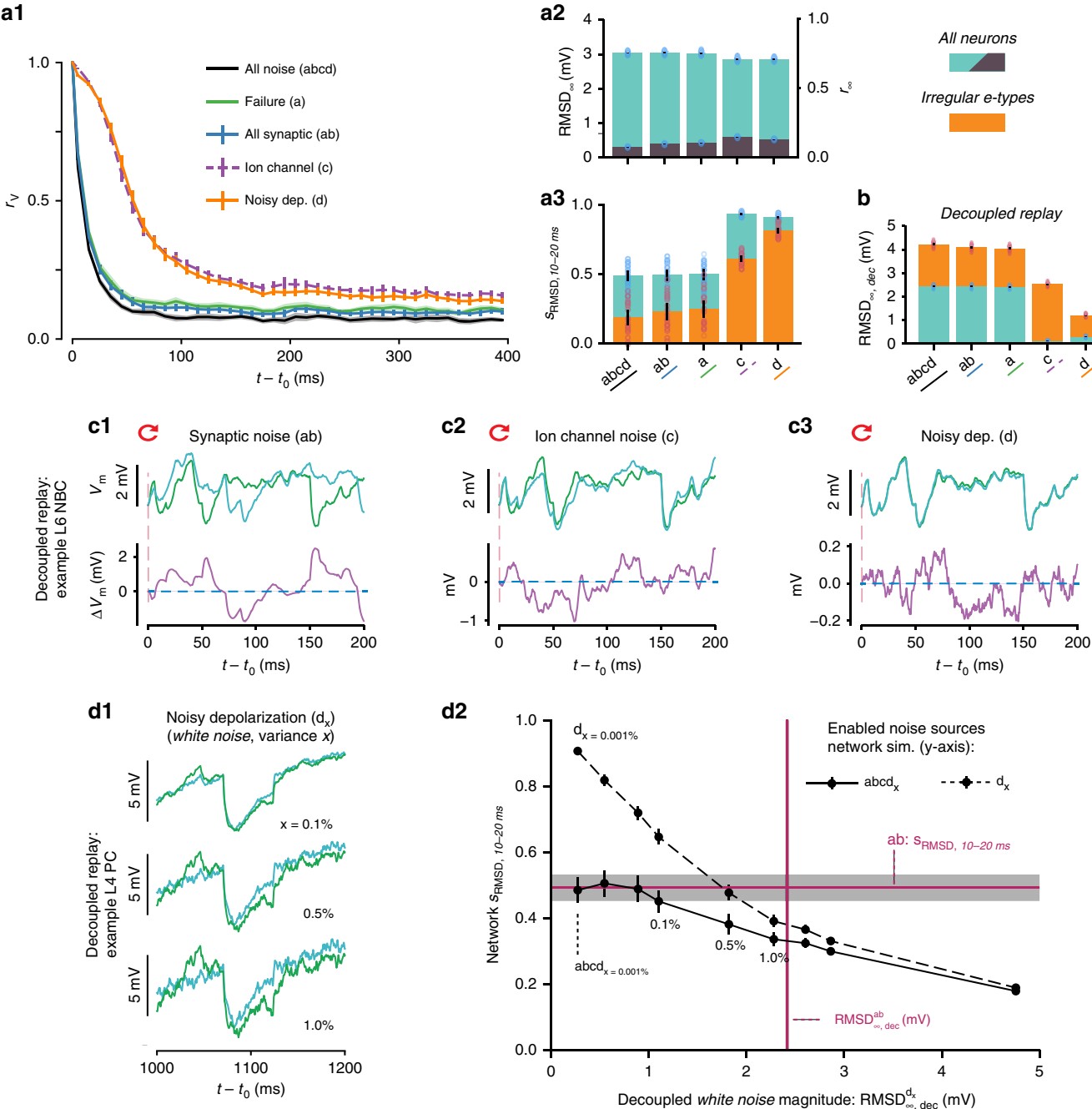

**Fig. 4** Synaptic noise dominates divergence. **a1** Time-course of correlation $r_V$ after resuming at $t_0$ from identical conditions with different noise sources enabled (*abcd*: 40 bases states; *a, ab, c, d*: 20 base states; mean ± 95% confidence interval). **a2** Steady-state root-mean square deviation RMSD$_\infty$ (cyan) and correlation $r_\infty$ (purple) with different noise sources enabled. Dots indicate individual trials; black error bars indicate 95% confidence interval. **a3** Similarity $s_{RMSD}$ at 10–20 ms with different noise sources enabled, for all neurons (cyan) and irregular e-types (orange). **b** Steady-state root-mean square deviation for decoupled simulations, RMSD$_{\infty,dec}$, for all neurons (cyan) and irregular e-types (orange) (20 saved base states). Irregular e-types: 1137 out of 31,346 neurons. **c** Decoupled replay simulations for a representative L6 NBC neuron, with somatic membrane potential differences between the two trials only due to synaptic noise (*ab*), ion-channel noise (*c*), or a noisy current injection (*d*). **d1** The effect of changing random seeds for the noisy depolarization only, for different noise strengths in a decoupled simulation. *x*: white noise variance as percentage of mean injected current. **d2** The decoupled steady-state membrane potential fluctuations RMSD$_{\infty,dec}^{d_x}$ evoked by different magnitudes of white noise without network dynamics, vs. the similarity $s_{RMSD}$ at 10–20 ms during network simulations when either turning on only the white noise depolarization (*d*) or all noise sources (*abcd*). Similarly, in purple, RMSD$_{\infty,dec}^{ab}$ for synaptic noise vs. the similarity at 10–20 ms when only turning on synaptic noise (*ab*). All error bars and shaded areas indicate 95% confidence intervals. Means for **d2** are based on ten base states

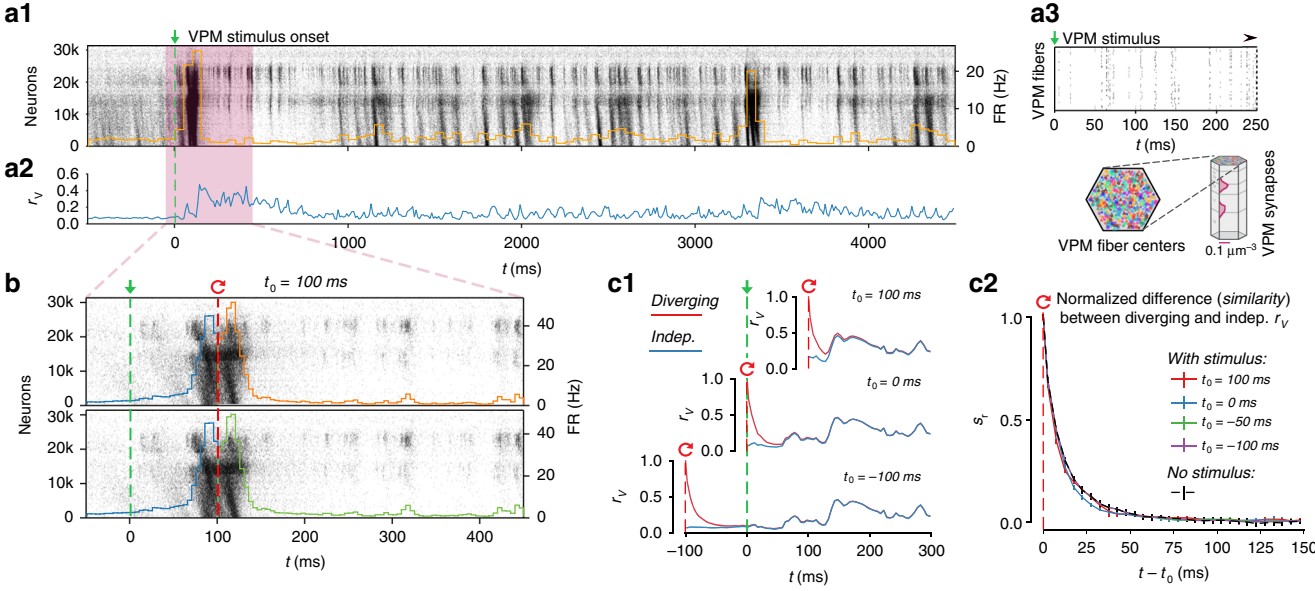

**Fig. 5** Rapid divergence of evoked, reliable activity. **a1** Population raster plot and population peristimulus time histogram (PSTH) for all 31,346 neurons in the microcircuit, during evoked activity with a thalamic (VPM) stimulus. Neurons are ordered according to cortical depth, with deep layers at the bottom and upper layers at the top, and each row representing the spikes of one neuron. For visibility, raster lines extend over dozens of rows for each neuron. **a2** Mean somatic membrane potential correlation $r_V$ between independent simulations of the same VPM stimulus (mean ± 95% confidence interval). **a3** Schematic of the VPM stimulus. Top: Raster plot spike times for the first 250 ms of the thalamic stimulus. Bottom: 310 VPM fiber centers are assigned 30 colors, and those with identical colors are provided with duplicate spike trains. The synapse density profile across layers for each fiber is shown to the right. **b** For $t < 100$, the top and bottom raster plots show the same simulation, whereas for $t > 100$, the raster plots depict two resumed simulations starting from the same saved state at $t_0 = 100$, using different random number seeds. **c1** Resuming from identical initial conditions at different times: during (top), at onset (middle), or before the stimulus (bottom). Mean $r_V$ between independent simulations (blue, as in **a2**), and mean $r_V$ between simulations starting from the same base state (red; mean ± 95% confidence interval). **c2** The similarity, $s_r$, defined as the difference between the $r_V$ of diverging and independent trials, normalized to lie between 1 (identical) and 0 (fully diverged) (mean ± 95% confidence interval). Means are based on 20 base states, *no stimulus* (spontaneous activity) on 40 as before

understand how the reliable responses emerge, we quantified the respective contribution of network propagation and cellular noise sources to variability. As before, we compared network simulations with decoupled replay simulations (Fig. 6a3). Unsurprisingly, $r_V(t)$ was much larger in the decoupled simulations (Fig. 6a1, black) than in the network simulations (Fig. 6a1, red; same as Fig. 5a2). However, the difference between the two was always smaller during evoked activity (Fig. 6a2, after 0 ms) than during spontaneous activity (Fig. 6a2, before 0 ms). This suggests that network dynamics play a reduced role in generating variability during evoked activity.

Indeed, when we focused on individual neurons (Fig. 6b), we saw that that the difference between network and decoupled $r_V(t)$ collapsed to zero at times (Fig. 6c). This is in stark contrast to spontaneous activity, where there is always a large difference between network and decoupled $r_V(t)$ (Supplementary Fig. 10a, c1). Hence, it appears that, in response to a stimulus, membrane potential variability due to network dynamics can intermittently be completely overcome, with remaining variability being solely due to cellular noise—at least for a sub-population of neurons in the network.

**Spike-time reliability amid noise and chaos**. How does the decreased membrane potential variability relate to spike-time reliability? Spike-timing is determined by a non-linear transformation of the somatic membrane potential. First, we observed that during spontaneous activity, the increase in membrane potential reliability in a decoupled replay does not directly translate into an increase in spike-time reliability (Supplementary Fig. 10b, c2). In fact, we found a small negative correlation (Supplementary Fig. 10b,

c3). However, during evoked activity, we observed from our example neurons that there are periods of reliable spiking where network variability can go to zero (Fig. 6c vs. 6d, top).

So far, we have analyzed the variability of spiking activity of neurons in terms of the Fano factor of their spike count (Fig. 1, see above). This measure quantifies average variability over relatively long time-windows and therefore cannot quantify the transient periods of reliability we observed. Therefore, we used a correlation-based measure of spike-time reliability, $r_{spike}$[34], to compare simulations with and without network dynamics for a population of neurons during evoked activity (Fig. 6d, e1). Contrary to the Fano factor, this measure is affected by the precise timing of spikes in smaller time windows. We observed that removing network dynamics only moderately increased spike-time reliability (Fig. 6e1, red vs. solid black line). In fact, increases in reliability were small for all neurons (Fig. 6e2, solid black line). In stark contrast to the spontaneous case (Supplementary Fig. 10c2), a small population of neurons in the evoked network simulations achieved close to perfect spike-time reliabilities (Fig. 6e1). As expected, most of the noise effects could be explained by synaptic noise alone (Fig. 6e1, 2, dotted black line).

We conclude that during spontaneous activity, the reliability of spike generation across timescales is directly, and severely constrained by synaptic noise, even without amplification through network dynamics. However, external stimuli can sparsely and transiently overcome chaotic network dynamics for sub-populations of neurons, with a residual variability—caused by synaptic noise—that is much smaller than during spontaneous activity.

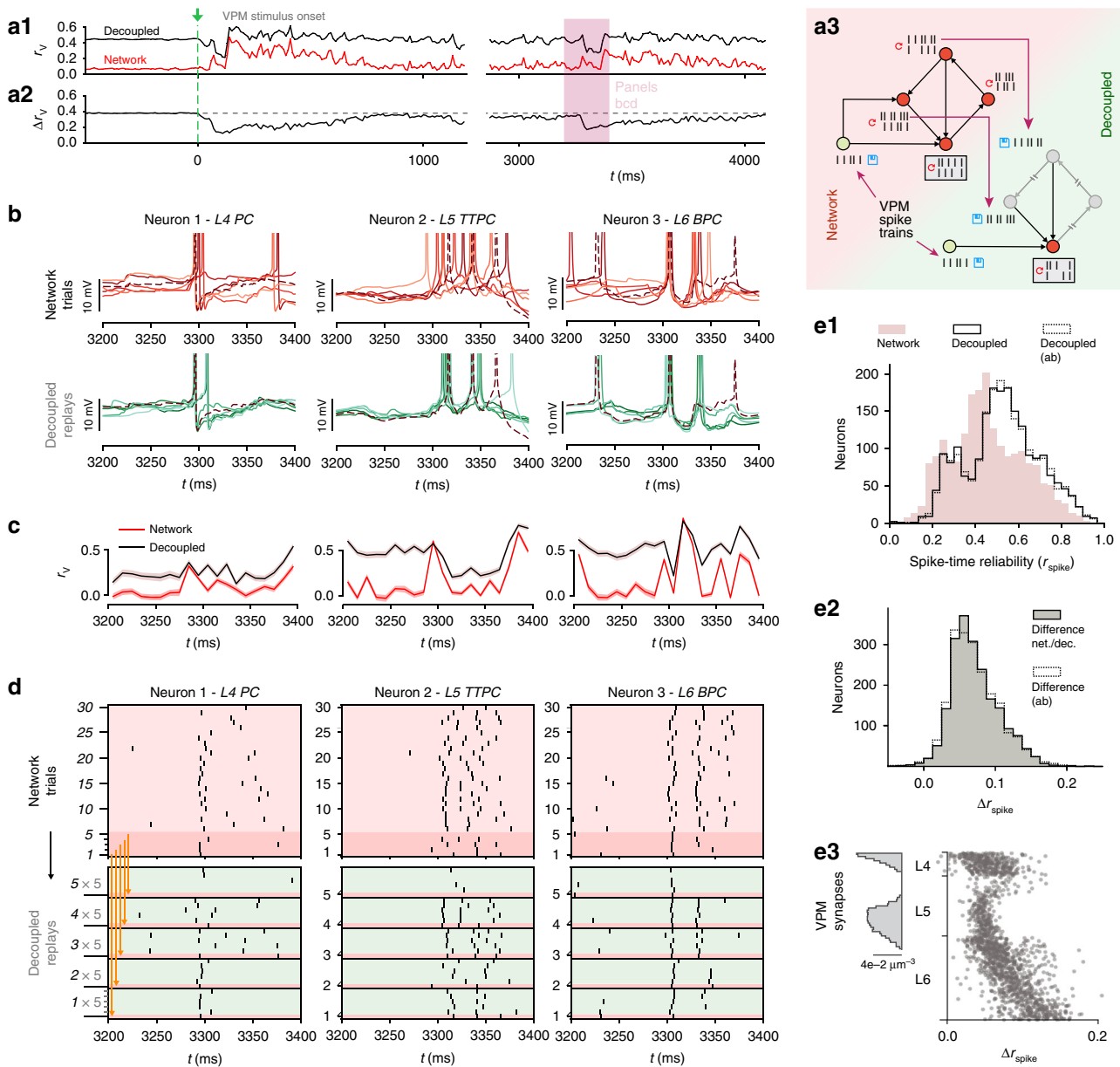

**Fig. 6** Spike-time reliability amid noise and chaos. **a1** Mean somatic membrane potential correlation, $r_V$, between independent simulations, and between decoupled replays of those simulations (network simulation identical to Fig. 5a2). **a2** Difference in $r_V$ for decoupled and network simulations. **a3** Schematic of network and decoupled replay simulation paradigms, including thalamic input. **b** Somatic membrane potentials ($V_m$) of three representative neurons for the time interval highlighted by the red box in **a**. Top: during six independent trials. Bottom: five decoupled replay trials (green) with the same presynaptic input as during the original network simulation trial (red), but with different random seeds. **c** Network and decoupled $r_V$ as in **a**, but only for the three sample neurons in **b**. **d** Top: Raster plot of spike times of the same three example neurons as in **b**, during 30 independent trials of evoked activity. Bottom: Decoupled replay trials (green) of the same input received during 5 of the 30 original trials (dark red). **e1** Mean spike-time reliability $r_{spike}$ of 2024 pyramidal neurons from layers 4, 5, and 6 between independent network simulations, and between decoupled replay simulations with identical presynaptic inputs. **e2** Difference between $r_{spike}$ of decoupled and replayed simulations. **e3** Difference between $r_{spike}$ of decoupled and replayed simulations vs. position of somata across layers 4, 5, and 6 of microcircuit (1675 neurons)

**High reliability requires recurrent cortical connectivity**. It is conceivable that the spike-time reliability we observed was simply a result of direct feed-forward thalamic input[35]. Indeed, when we looked at changes in reliability without network dynamics, the strongest increase in reliability was in neurons at the bottom of layer six that receive comparatively little direct VPM input (Fig. 6e3). On the other hand, the VPM input was weak compared to the recurrent connectivity, making up only 7% of the connections onto neurons in layer 4, 4% for layer 5, and <3% for layer 6. To test whether the intermittent suppression of chaotic dynamics is simply an effect of the feed-forward input, we compared simulations of single cells with network simulations. To this end, we designed a new simulation paradigm similar to our previous decoupled replay, where each neuron received a combination of replayed presynaptic inputs from a simulation of spontaneous activity and from the direct feed-forward VPM input it received in the evoked network simulations (Fig. 7a1). That is, each neuron receives input as in a spontaneous activity trial through its recurrent synaptic contacts, and input as in an evoked trial through its feed-forward synaptic contacts.

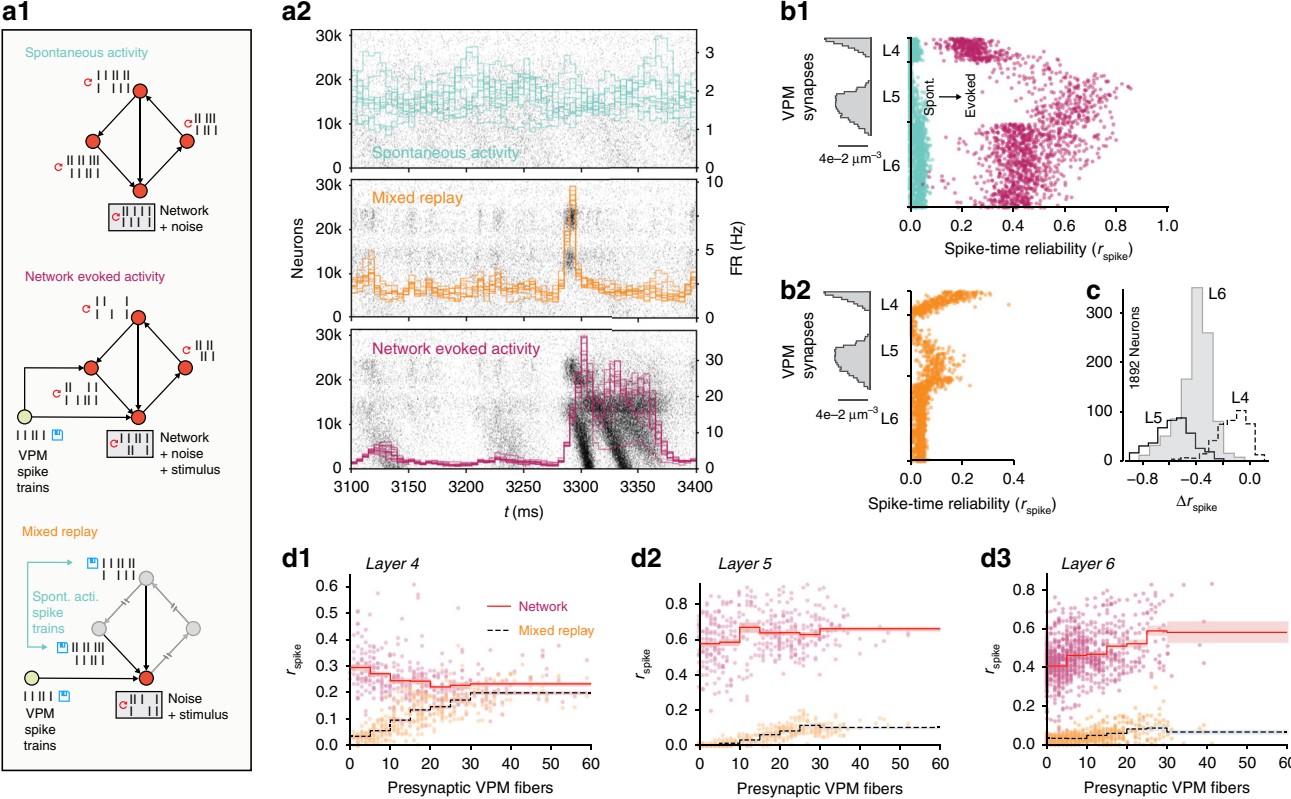

**Fig. 7** High reliability requires recurrent cortical connectivity. **a1** Overview of three simulation paradigms: *spontaneous activity*, *network evoked activity* (with network propagation intact and VPM input), and *mixed replay* (with network propagation replaced by replays of spontaneous activity spike trains, and VPM input). **a2** Examples of population spiking activity during the three simulation paradigms. **b1** Spike-time reliability, $r_{spike}$, during spontaneous (blue) and evoked (purple) activity for 1675 excitatory neurons in the center of layers 4, 5, and 6. **b2** Spike-time reliability, $r_{spike}$, during a mixed replay with VPM input but with network propagation disabled for the same neurons as in **b1**. **c** Difference in $r_{spike}$ between evoked activity with and without network propagation for 1892 excitatory neurons in the center of layers 4, 5, and 6 (same for **d1–3**). **d1** The number of presynaptic VPM fibers from which each neuron receives input vs. $r_{spike}$ in evoked simulations with (*network*) and without (*mixed replay*) network propagation. All reliabilities per neuron (points) are based on 30 trials. Mean of neurons per bin indicated by line; shaded area depicts standard error of mean of neurons in bin. Bins: 0–4, 5–9, 10–14, 15–19, 20–24, 25–29, 30+ VPM fibers

In this mixed replay paradigm, the population response was much weaker (Fig. 7a2). While in simulations of evoked activity, all neurons showed higher reliability than in simulations of spontaneous activity (Fig. 7b1), in the mixed replay, the only cells that showed increased reliability were those close to the VPM synapses (Fig. 7b2). Furthermore, the only neurons to display similar reliability, with and without recurrent network propagation, were a small group in layer 4 (Fig. 7c). Taken together, these findings suggest that feed-forward VPM input alone is not enough to make the majority of neurons spike reliably.

To test this hypothesis, we compared the reliability between the two simulation paradigms to the number of presynaptic VPM fibers innervating each neuron (Fig. 7d1–3). We can see that neurons in layer 4 that receive little direct VPM input responded more reliably with the network enabled than neurons that receive a lot of VPM input in the mixed replay case (Fig. 7d1). Neurons in layers 5 and 6 were more reliable in mixed replay when they had more presynaptic VPM connections. However, this reliability increases drastically when network dynamics are enabled (Fig. 7d2, 3). We conclude that the reliable spiking observed in response to VPM inputs is propagated and amplified by recurrent cortical connectivity.

**High reliability emerges near a critical EI-balance**. What mechanisms allow the recurrent cortical circuitry to respond so reliably? We have shown above that dynamics in the NMC-model

depend on the balance between excitatory and inhibitory activity (EI-balance). This balance can be altered by the extracellular calcium concentration ($[Ca^{2+}]_o$), which differentially modulates the effective strength of excitatory and inhibitory synapses. At $[Ca^{2+}]_o \approx 1.25$ mM, the circuit is in a state where it exhibits several properties of in vivo dynamics[17] (Figs. 8a2 and 2a1). For lower $[Ca^{2+}]_o$, activity becomes more and more asynchronous (Fig. 8a1), for higher $[Ca^{2+}]_o$, activity reaches a critical point and abruptly transitions to synchronous, regenerative (supercritical) behavior (Figs. 8a3 and 2a2).

To understand how this affects spike-time reliability, we repeated 30 trials of the thalamic stimulus at eight different levels of $[Ca^{2+}]_o$ (Fig. 8a, b). As we go from asynchronous to synchronous dynamics, the response properties visibly change (Fig. 8a) and spiking becomes more reliable (Fig. 8c). As we approach the state at $[Ca^{2+}]_o = 1.25$ mM, reliability increases sharply (Fig. 8c), whereas the overall EI-balance increases gradually (Fig. 8d). As we transition to supercritical regenerative, synchronous behavior the reliability begins to plateau. At the same time, the population response becomes erratic, with all-or-nothing network bursts (Fig. 8b and Supplementary Fig. 11). We previously showed that stimulus discriminatory power breaks down in this supercritical regime, as it does far into the asynchronous regime[17].

We conclude that spike-time reliability in the microcircuit is adaptive: any neuromodulator that differentially targets inhibitory

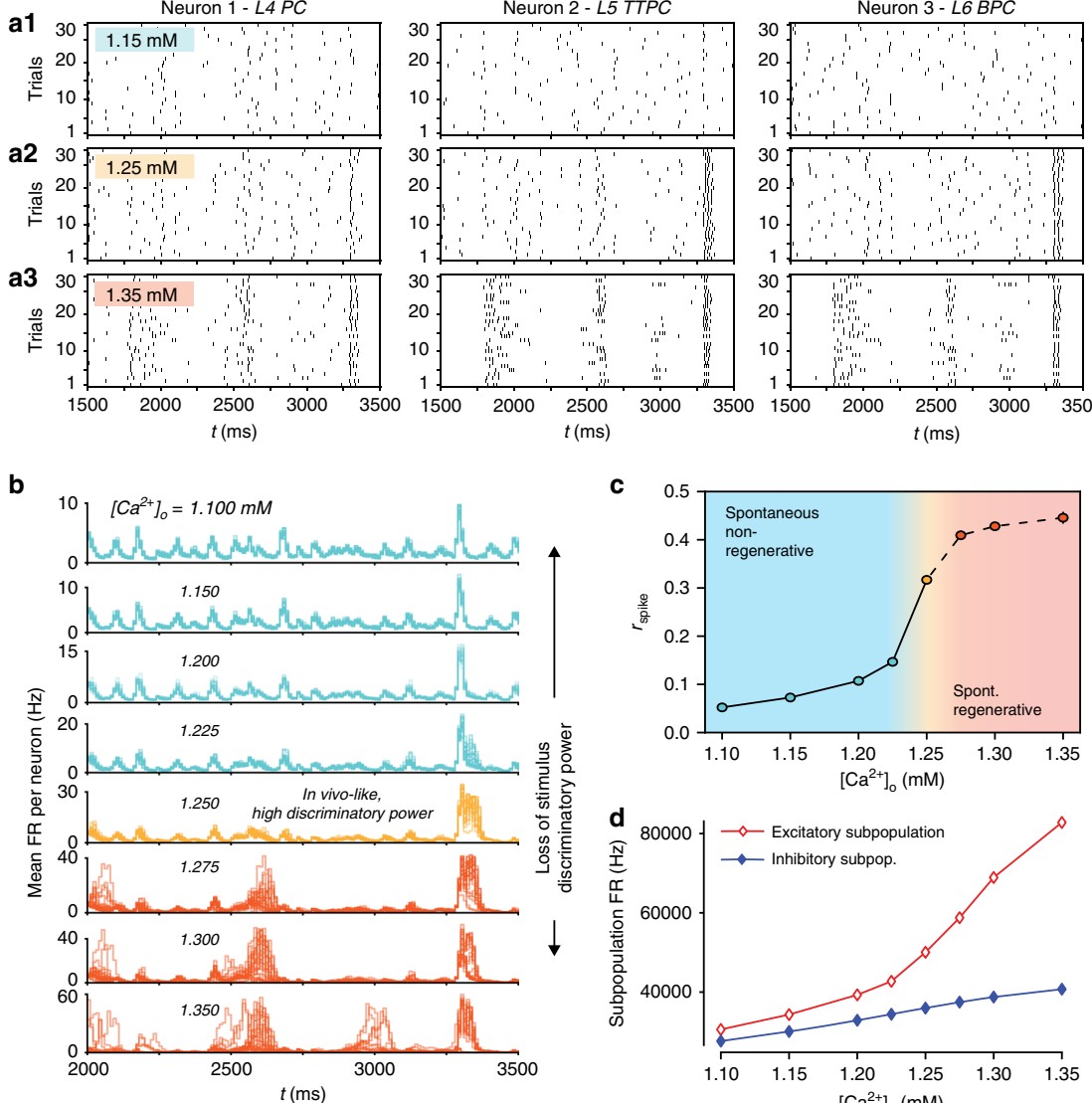

**Fig. 8 High reliability emerges near a critical EI-balance. a1–3** Raster plot of spike times of the same three example neurons as in Fig. 6d, during 30 independent trials of evoked activity, for three different extracellular calcium concentrations ($[Ca^{2+}]_o$ = 1.15, 1.25, 1.35 mM). **b** Population firing rates for all 30 trials for eight different extracellular calcium concentrations. Loss of stimulus discriminatory power away from $[Ca^{2+}]_o$ = 1.25 mM as observed by Markram et al.[17]. **c** Mean reliability of 2024 pyramidal neurons from layers 4, 5, and 6 during the evoked response vs. extracellular calcium concentration (mean of 30 trials ± 95% confidence interval). **d** Total population firing rates for excitatory and inhibitory subpopulations in the whole microcircuit during the evoked response

and excitatory synapses could adapt the response reliability in the microcircuit according to computational tasks by fine-tuning the global EI-balance.

## Discussion

In the present study, we used a biologically constrained model of a prototypical neocortical microcircuit[17] to estimate the internally generated variability of local neocortical activity (Figs. 1–4) and explored the implications for reliable stimulus encoding (Figs. 5–8). We found that cortical circuitry supports millisecond-precision spike-time reliability amid highly variable, chaotic network activity. This resolves a long-standing question: Is the cortex too noisy for the precise timing of a spike to matter[8,12,21,23]? Put simply, if spiking is unreliable, information must be coded by firing rates estimated in populations of neurons[21,23], whereas if it is reliable, precise spike timing of single neurons could contain

significant information[8,12]. Here, we demonstrated that cortical circuitry naturally supports both regimes.

The experimental manipulations required to untangle the noise sources in the brain, and evaluate their impact on spike reliability, are impossible to perform in vitro or in vivo. Using the NMC-model, we were able to perform a series of simulation-based manipulations where we systematically added and removed noise sources to quantify their impacts. These manipulations yielded several novel insights.

First, we found that spontaneous activity in cortical circuitry is intrinsically variable in terms of rapidly diverging activity trajectories, both at the single neuron and population level (Figs. 1 and 2). While some of the effects of cellular noise sources on variability had been studied in single biophysical Hodgkin–Huxley type neuron models[20,35–37], this is the first estimate of internally generated variability in an integrated, biologically constrained model of a cortical circuit.

Our second insight was that stochastic synaptic transmission is amplified by chaotic network dynamics to drive a rapid chaotic divergence of the network, resulting in the above-mentioned variability (Figs. 3 and 4). Chaotic network dynamics without synaptic noise have been extensively studied[21–23], and it has been suggested that synaptic noise can generate high neural variability in postsynaptic neurons[38] and recurrent networks[18].

In spite of this rapid divergence of activity, variability in terms of spike counts was below the expected value for Poisson-like spiking[18] (Fig. 1f). Our model predicts that Poisson-like spike-count variability in vivo is not internally generated, but rather reflects non-local inputs (see Supplementary Discussion, Supplementary Figs. 12 and 13), in line with recent in vivo experiments[39]. The relatively stable spike counts during the chaotic, divergent spontaneous activity likely arise from a combination of spike frequency adaptation mechanisms[24,40,41] and synaptic filtering[25]. This is in contrast to previous findings, which suggested that synaptic noise and recurrent network dynamics lead to Poisson-like spike-count variability[18].

The third insight was that comparatively weak thalamocortical input could switch the network to a highly reliable spiking regime (Figs. 5 and 6). This confirms results of deterministic models[42] in the presence of cellular noise and weak thalamic inputs, and explains how patterns of activity generated by cortical circuitry in response to sensory stimuli can often have millisecond spike-time precision[43,44].

The fourth insight is the mechanism for this dichotomous behavior. We determined that the recurrent network architecture causes both the amplification of synaptic noise during spontaneous activity, and the reliable response in the presence of input (Fig. 7). The critical role of the recurrent network stands in contrast to previous modeling work, which showed that relatively few synchronous thalamic inputs maximize reliability in single neurons in cat visual cortex[35]. However, this study likely overestimated synaptic reliability—synaptic release probabilities are lower in vivo than in vitro, both in general[16] and in this specific pathway[45]. We found that the reliability of the response strongly depended on the overall EI-balance in the network (Fig. 8). The response reliability rapidly increases towards a just subcritical dynamical state where the microcircuit reproduces several in vivo findings and neurons have maximum discriminatory power to different stimuli[17].

The exact mechanism for this triggering of reliable spiking, and the means by which signals are reliably propagated through the circuitry amid variable activity remain a subject for future investigation. In a first step towards a characterization, we found that synchronous inputs that arrive within several milliseconds are well suited to elicit reliable responses (Supplementary Fig. 14), in line with previous predictions for single cells[35]. One possible explanation for the propagation of reliable activity is that certain connectivity motifs could amplify reliability through redundant connectivity, such as common neighbor motifs[46] and high-dimensional cliques that shape spike correlations between neurons[32]. In fact, we found that neurons with high in-degrees were more reliable (Supplementary Fig. 15a), and that neurons are less reliable the fewer higher-dimensional cliques[32] they are part of (at similar in-degree, Supplementary Fig. 15b). Dendritic non-linearities, such as N-methyl-D-aspartate (NMDA)-mediated plateau potentials evoked by clustered synaptic inputs onto the dendritic tree could also play an important role[47,48].

While the NMC-model is one of the most detailed models of neocortical circuitry to date, several biological details are lacking. Multivesicular release might decrease synaptic variability (see Supplementary Discussion and Supplementary Fig. 16)[49,50]. In terms of noise sources, the most important lacking detail is ion-channel noise[13,36,51], which could increase variability of spike

timing by up to several milliseconds in long axons[52] (see Supplementary Discussion). There are other internal mechanisms not yet included in the NMC-model such as gap junctions, intra-circuit neuromodulation[53] or active information transfer from glia to neurons[54,55], whose contributions to variability within cortical circuits are as yet poorly understood. However, for these mechanisms to contribute significantly as additional noise sources beyond synaptic noise, they would have to cause somatic membrane potential fluctuations on the order of 1 mV (Fig. 4).

This study provides a data-constrained biophysical framework that supports theories of cortical coding along a spectrum—from population firing rates to reliable individual spike times. This study does not claim that cortex generally employs codes that rely on individual spike times, only that it is principally capable of such codes. Even a highly reliable cortex might be variable due to computational strategies that are intrinsically variable, such as sub-optimal inference[56], or due to overcomplete representation of inputs, with distributions of spike patterns encoding the same stimulus[57]. Encoding strategies might further be adjusted according to computational needs by fine-tuning of the network near criticality[58], for example, due to neuromodulation that shifts the balance between excitation and inhibition and with it spike-time reliability.

## Methods

**Model of neocortical microcircuitry (NMC-model)**. Simulations of electrical activity were performed on a previously published model of a neocortical microcircuit based on data from two-week old rats. Reconstruction and simulation methods are described extensively by Markram et al.[17]. In our study, we used a microcircuit consisting of 31,346 biophysical Hodgkin–Huxley NEURON models and around 7.8 million connections forming roughly 36.4 million synapses. Synaptic connectivity between 55 distinct morphological types of neurons (*m-types*) was predicted algorithmically by integrating anatomical data, such as layer-dependent cell type densities, morphologies, and bouton densities, to generate a wiring diagram[59] with highly heterogeneous connectivity[32,60,61]. Consequently, the NMC-model exhibits a naturally emerging structural and functional EI-balance[60], without relying on assumptions about the exact level of coupling between excitatory and inhibitory currents. The densities of ion channels on morphologically detailed neuron models were optimized to reproduce the behavior of different electrical neuron types (*e-types*) as recorded in vitro[62].

The NMC-model contains three types of biological noise sources, all of which are required to replicate neuronal responses to paired recordings and current injections in vitro. Each of the 36 million synapses in the model incorporates stochastic models of vesicle release with biologically constrained variability, which display both *failure* of vesicle release (*a*) and *spontaneous release* (*b*). The neuron types that exhibit irregular firing behavior (1137 neurons) also contain models of *stochastic potassium channels* (*c*), which induce irregular firing in response to constant current injections in vitro. A fourth, tunable noise source consisted of a noisy current (*d*) injected at the soma of each of the 31,346 neurons in the model, used to account for other putative sources of depolarization in vivo. We maintained the magnitude of this generic noise far below the magnitude of the experimentally constrained noise sources, using it later for sensitivity analysis (variance of 0.001% of the mean injected current per neuron, unless stated otherwise). In our initial experiments, all noise sources are thus intrinsic to the microcircuit, with all variability generated internally.

We also used a larger mesocircuit comprising seven microcircuits (mean of 36.5 million synapses per circuit), with no boundaries between the peripheral circuits and the original microcircuit in the center. Simulations were run on a BlueGene/Q supercomputer (BlueBrain IV) and an HPE SGI 8600 supercomputer (BlueBrain V). NEURON[63] models and the connectome are available online at bbp.epfl.ch/nmc-portal[64].

**Simulation of spontaneous activity**. Neurons were depolarized with a somatic current injection, with currents expressed as a percent of first spike threshold for each neuron, to mimic, for example, the effect of depolarization due to missing neuromodulators. Release probabilities for all synapses were modulated according to the extracellular calcium concentration found in vivo, leading to substantially lower reliability than in vitro[16]. As described by Markram et al.[17], the $U_{SE}$ parameter for synaptic transmission was modulated differentially as a function of extracellular calcium concentration ($[Ca^{2+}]_o$). Excitatory synapses are more strongly affected by $[Ca^{2+}]_o$ changes than inhibitory synapses, whereby an increase in the concentration of $[Ca^{2+}]_o$ shifts the EI-balance of the network in favor of excitation. It was previously shown that such changes in $[Ca^{2+}]_o$ induce a sharp transition in network activity, from asynchronous to regenerative synchronous activity. This transition occurs around a critical point just above $[Ca^{2+}]_o = 1.25$ mM, with activity below

this point being subcritical and activity above this point being supercritical. With mean injected currents at around 100% of first spike threshold and $[Ca^{2+}]_o$ at 1.25 mM, the microcircuit exhibits spontaneous activity that reproduces several properties of in vivo spontaneous and evoked activity[17].

**Simulation of evoked activity.** The microcircuit is innervated by 310 (virtual) thalamic fibers[17]. In vivo spike train recordings from 30 VPM neurons were randomly assigned to the 310 fibers, to achieve varying degrees of naturalistic synchronous thalamic inputs. Spike trains were recorded during replayed whisker motion in anesthetized rats[33]. Full methods are described in Reimann et al.[32]. A variable version of the naturalistic input used in vivo spike train recordings of the same 30 VPM neurons during 30 trials of the same replayed whisker motion[33]. Another stimulus consisted of synchronous spikes at the 60 central thalamic fibers, with all 60 virtual thalamic neurons firing simultaneously, to approximate a whisker "flick" (as in Markram et al.[17]).

**Save-resume.** After running a simulation for some amount of biological time, the final states of all variables in the system were written to disk using NEURON's *SaveState* class. For large-scale simulations, this required the various processes to coordinate how much data each needed to write, so that each rank could then seek the appropriate file offset and together write in parallel without interfering with the others. After restoring a simulation, the user could specify new random seeds (see below).

**Random numbers.** In our simulations, we used random number generators (RNGs) to model all stochastic processes: noisy current injection, stochastic ion channels, probabilistic release of neurotransmitters and generation of spontaneous release events. Each synapse had two RNGs. One was used to determine vesicle release on the arrival of an action potential. The other determined the spontaneous release signal. Similarly, each stochastic $K^+$-channel model had an RNG determining voltage-dependent opening and closing times. Finally, the white noise process underlying the noisy depolarization was determined by one RNG per neuron. By using different random seeds to initialize the RNGs, we obtained different sequences of random numbers, and consequently different but equally valid simulation outcomes. In earlier versions of the NEURON microcircuit simulation software, the user was given only a single random seed parameter with which to alter the random number streams generated by all RNGs. We added the option to separately change random seeds for RNGs for a specific type of stochastic component. For example, "IonChannelSeed <value>" allows the specification of a seed, which is only given to the RNGs used by ion-channel instances.

**Stochastic synapses.** The synapse models including parameters are described in detail in Markram et al.[17], and the models used can be found online at bbp.epfl.ch/nmc-portal. The model is based on two previous models[25,65]. In short, each synapse has a pool of readily releasable vesicles of size $n_{rrp}$, which are in one of two states: recovered or depleted. Upon action potential arrival at the synapse, each recovered vesicle stochastically releases with dynamic probability $U(t)$. The probability of vesicle release $U(t)$ is dynamic, to implement synaptic facilitation, and is governed by an event-based equation:

$$U(t) = U(t_{syn}) \cdot e^{-\frac{t-t_{syn}}{\tau_{fac}}} + U_{SE} \cdot \left(1 - U(t_{syn}) \cdot e^{-\frac{t-t_{syn}}{\tau_{fac}}}\right), \quad (1)$$

where $U_{SE}$ is the release probability of a synapse that has not been activated in a long time, $t_{syn}$ is the time of arrival of the last presynaptic spike at the synapse, and $\tau_{fac}$ is the facilitation time constant. For each released vesicle, postsynaptic AMPAR and NMDAR models are activated with a conductance $g_{max}/n_{rrp}$ where $g_{max}$ is the maximal postsynaptic conductance. After successful vesicle release, the vesicle location is in a depleted state in which it has no vesicle to release. The transition from the depleted state back to the recovered state is governed by a Poisson process, according to a survival function:

$$P_{surv}(t) = e^{-(t-t_{syn})/\tau_{dep}}, \quad (2)$$

where $P_{surv}(t)$ is the probability of remaining in the depleted state in the interval $[t_{syn}, t]$, and $\tau_{dep}$ is the depression time constant. The univesicular case ($n_{rrp} = 1$) is modeled, unless stated otherwise.

A second stochastic process is used to generate event times for spontaneous "miniature" postsynaptic potentials. Spontaneous release is modeled as an independent Poisson process with a rate $\lambda_{spont}$ at each synapse. When the synapse receives the signal for a spontaneous release event, it is treated as a presynaptic action potential.

**Deterministic synapse model.** The deterministic synapse model is implemented as previously described[25]. In this formulation, $U_{SE}(t)$ is interpreted as the fraction of consumed resources, rather than a release probability. Each release event activates a fraction of postsynaptic conductance proportional to $U_{SE}(t) \cdot R(t)$, where $R(t)$ is the fraction of vesicles in the recovered state. These two state variables are

governed by the following equations[25]:

$$\frac{dR}{dt} = \frac{(1-R)}{\tau_{rec}} - U_{SE} \cdot R \cdot \delta(t - t_{syn}), \quad (3)$$

$$\frac{dU_{SE}}{dt} = -\frac{U_{SE}}{\tau_{facil}} + U1 \cdot (1 - U_{SE}) \cdot \delta(t - t_{syn}), \quad (4)$$

where $\tau_{rec}$ and $\tau_{facil}$ are the recovery and facilitation relaxation time constants, $U_{SE}$ is a dynamic variable that increases by an amount determined by U1 for each presynaptic spike (note that U1 is equivalent to $U_{SE}$ in the stochastic model), and $t_{syn}$ is the time of arrival of presynaptic spikes at the synapse. The deterministic models implemented in this way are equivalent to their stochastic (multivesicular) counterparts in the limit as $n_{rrp} \rightarrow \infty$.

**Stochastic ion channels.** In some interneuron models, a potassium channel type with a stochastic implementation was added using previously described methods[17,20,36,66] to model ion-channel noise. The full model is available online at bbp.epfl.ch/nmc-portal. In brief, instead of a mean field model, the equations used explicitly track the number of channels in a certain state and allow these numbers to evolve stochastically, according to a binomial distribution, with the probability of transition between states computed according to the deterministic rate functions $\alpha$ and $\beta$:

$$\text{Open} \underset{\beta}{\overset{\alpha}{\rightleftharpoons}} \text{Closed}$$

**Single spike injection.** We injected single spikes in 20 different layer 4 pyramidal neurons by replaying (see below) an additional spike event in one neuron per simulation. Thus, there were no shifted or missing spikes, as may occur when injecting a spike in vivo. The spike was injected 0.1 ms after resuming the simulation from identical initial conditions.

**Step-pulse perturbation.** We applied a microscopic current step-pulse to all neurons at their soma 0.1 ms after resuming the simulation (duration: 0.1 ms, amplitude: 1 pA). The current was chosen to have an almost negligible effect on individual neurons, and was near the limit of the NEURON integrator. On average, $108 \pm 8$ neurons out of 31,346 neurons had any changes in their spike times (mean of 19 trials ± STD). The majority of the shifted spikes were shifted by <0.05 ms (59.1%: <0.05; 33.1%: <1 ms; 5.5%: <20 ms; 1.8%: <100 s; 0.5%: <1 s). Finally, $3 \pm 2$ neurons had extra or missing spikes. The median first occurrence of an extra or missing spike was at 257 ms (min: 11 ms, max: 946 ms after resuming).

**Decoupled replay.** When resuming a simulation at $t_0$, we decoupled all connections by setting the connection weights to zero, ensuring that action potentials would be delivered to the synapses of postsynaptic neurons. At the same time, we started replaying action potential times from a previous resumed simulation, activating the synapses of postsynaptic neurons as if the presynaptic neuron had fired an action potential, but actually replaying presynaptic action potentials from the previous simulation. For computational reasons, spikes that had not been delivered at the save time $t_0$, were not delivered in the decoupled replay (meaning that a couple of presynaptic spikes per neuron may have been lost, leading to a slight underestimation of divergence).

**RMSD and correlation.** For each neuron $n$, we calculated the root-mean-square deviation $\text{RMSD}_V(n, k; t)$ of its somatic membrane potential between two trials in time bins of size $\Delta t$ starting from $t_0$:

$$\text{RMSD}_V(n, k; t) = \sqrt{\int_{t-\Delta t/2}^{t+\Delta t/2} [V_{m,1}(n, k; t') - V_{m,2}(n, k; t')]^2 dt'/\Delta t}, \quad (5)$$

where $V_{m,1}(n, k; t)$ and $V_{m,2}(n, k; t)$ denote the time series of somatic membrane potentials of neuron $n$ in the two respective trials resuming from the same base state $k$. We consequently defined the mean root-mean-square deviation of the microcircuit $\text{RMSD}_V(t)$ as the mean of $\text{RMSD}_V(n, k; t)$ over all base states ($K = 40$) and neurons ($N = 31,346$).

We analogously computed the linear correlation of somatic membrane potentials between two trials in time bins of size $\Delta t$, starting from $t_0$ (averaging over $t'$ in each time bin of size $\Delta t$):

$$r_V(n, k; t) = \frac{\text{cov}\left(V_{m,1}(n, k; t'), V_{m,2}(n, k; t')\right)}{\sigma\left(V_{m,1}(n, k; t')\right) \cdot \sigma\left(V_{m,2}(n, k; t')\right)}, \quad t - \frac{\Delta t}{2} < t' \leq t + \frac{\Delta t}{2} \quad (6)$$

All analysis was performed using custom scripts written in Python 2.7 using the *NumPy*, *matplotlib*, and *SciPy* libraries. Scripts were executed on a Linux cluster connected to the same IBM GPFS file system that the simulation output was

written to. Root-mean-square deviation $RMSD_V$ and correlation $r_V$ were implemented with *NumPy*.

**Similarity**. The similarity measure $s(t)$ was defined as the normalized difference between diverging $r_V(t)$ (or $RMSD_V(t)$), and steady-state $r_V(t)$ (or $RMSD_V(t)$). The steady-state value was defined as the continuous $r_{V,\text{shuffle}}(t)$ computed by shuffling the soma voltages between simulation trials, so that instead of 40 deviating pairs of trajectories, we compared 40 independent pairs of trajectories. Alternatively, we defined it as the mean stationary, fully deviated $r_\infty$ for $t > 1000$ ms after resuming from identical initial conditions.

**Firing rate**. Firing rate was defined as the average number of spikes in a time interval of size $\Delta t$, divided by $\Delta t$ ($\Delta t = 10$ ms, unless stated otherwise).

**Neuron selection**. We selected all excitatory neurons in layers 4, 5, and 6 that belonged to the 30 minicolumns (out of 310 in total) in the center of microcircuit ($n = 2024$). The analysis was restricted to neurons that spiked at least once in each of the compared simulation paradigms.

**Spike-time reliability**. Spike-time reliability was measured using a correlation-based measure first proposed by Schreiber et al.[34]. Briefly, the spike times of each neuron in each trial were convolved with a Gaussian kernel of width $\sigma_s = 5$ ms to yield filtered signals $s(n, k; t)$ for each neuron $n$ and each trial $k$ ($\Delta t_s = 1$ ms). The spike-time reliability for each neuron was then defined as the mean inner product between pairs of signals divided by their magnitude:

$$r_{\text{spike}}(n) = \frac{2}{K(K-1)} \sum_{k \neq l} \frac{s(n,k;t) \cdot s(n,l;t)}{|s(n,k;t)| \cdot |s(n,l;t)|},$$ ($K = 30$; independent trials). Decoupled replay: there are $M = 5$ replays of each of the $K = 30$ trials, and thus

$$r_{\text{spike}}(n) = \frac{2}{KM(M-1)} \sum_m \sum_{k \neq l} \frac{s_m(n,k;t) \cdot s_m(n,l;t)}{|s_m(n,k;t)| \cdot |s_m(n,l;t)|}.$$

**Errors and statistical tests**. Error bars and shaded areas indicate 95%-confidence intervals (CI), unless stated otherwise. $t$-based CIs ($n = 20$; or $n = 40$ if stated) were computed using *scipy.stats.sem* and *scipy.stats.t.ppf* to compute $p$-values from the CIs (one-sided). Errors for fit parameters, obtained with *scipy.optimize.curve_fit*, are given as the square-root of the variance of the parameter estimate.

**Reporting summary**. Further information on research design is available in the Nature Research Reporting Summary linked to this article.

## Data availability

NEURON models, microcircuit information, and the connectome are available for download at https://bbp.epfl.ch/nmc-portal/downloads. The integrated microcircuit model is available upon reasonable request. Output spike times and output somatic membrane potentials are available upon reasonable request.

## Code availability

Software used for visualization of neurons in Fig. 1 is available at https://github.com/BlueBrain/RTNeuron. The NEURON simulation environment is available at https://www.neuron.yale.edu/neuron/. The custom-written Python analysis and figure generation scripts are available at https://github.com/maxnolte/deciphering_variability.

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

## Acknowledgements
We thank Giuseppe Chindemi, Srikanth Ramaswamy, and Werner Van Geit for help with synapse and ion-channel models, and the rest of the Blue Brain team for developing and maintaining the microcircuit model and computational infrastructure. We thank Taylor Newton, Madineh Sedigh-Sarvestani, Richard Walker, and Mickey London for discussions and critical comments on the manuscript. We thank Oren Amsalem, Idan Segev, Wulfram Gerstner, and Alexandre Pouget for helpful discussions. This study was supported by funding to the Blue Brain Project, a research center of the École polytechnique fédérale de Lausanne, from the Swiss government's ETH Board of the Swiss Federal Institutes of Technology.

## Author contributions
Conceptualization: M.N., M.R., H.M., E.M.; Methodology: M.N., M.R., H.M., E.M.; Software: M.N., J.K.; Validation: M.N., J.K.; Investigation: M.N; Visualization: M.N.; Writing—Original draft: M.N., M.R.; Writing—Review and editing: M.N., M.R., H.M., E.M.; Supervision: H.M., E.M.; Funding acquisition: H.M.

## Additional information

**Competing interests:** The authors declare no competing interests.

