## [Peer Review File · Nature Communications]

Editorial Note: This manuscript has been previously reviewed at another journal that is not operating a transparent peer review scheme. This document only contains reviewer comments and rebuttal letters for versions considered at Nature Communications .

Reviewers' Comments:

Reviewer #1:

Remarks to the Author:

The authors have responded appropriately to all my comments. In particular, adding the details of the models have clarified some of the issues that I had.

One minor comment is that the stochastic synaptic model resembles one studied previously:
<https://www.sciencedirect.com/science/article/pii/S0925231204000578#BIB9>

Reviewer #2:

Remarks to the Author:

I regret that the authors have chosen to argue against my comments rather than address them more directly. The manuscript presents an impressive amount of work, yet in the present format the results are mostly descriptive. A direct comparison between the analyses of simulations of the large scale model, and identical analyses in simplified, classical E-I networks would have been extremely valuable, and certainly appropriate for Nature Comm. The authors argue that this comparison is done in a different study, but I would have been glad to see it here.

However an important concern that has emerged in response to the comments of Reviewer 1 is that the spiking statistics in the large scale model are actually very far from Poisson, and seem very regular (it would have been nice to have more information about the statistics beyond Fano factors). This really puts into question what kind of dynamical regime the simulations are exhibiting, and how that relates to the cortex. The authors first argue that the activity in the cortex is not actually Poisson, which is certainly true, but the obtained Fano factors seem very small in comparison to any cortical data recorded in vivo at rest (cf e.g. Churchland et al 2010 for an overview that includes V1). The authors then argue that Poisson variability is not generated by recurrent connectivity but by external inputs. Now that is a strong and certainly controversial statement that would be a major result if it could be strongly substantiated (the authors show a bit of evidence for it in Fig S14, but there they show only a transient increase of Fano Factors, quite different from Churchland et al 2010). In any case, the authors seem to admit that as they do not have variable external inputs, in the main simulations their network is not in a regime that resembles the cortex, and this calls into question the relevance of the reported observations.

Response to the reviewers' comments:

We would like to thank the reviewers for their reading of the revised manuscript, and for their constructive feedback. We revised the manuscript to address their new comments, as described point-by-point below.

In response to Reviewer 2's comments we performed a new set of simulations that resulted in a new **Supplementary Fig. 15** (we renamed old **Supplementary Fig. 15** to **16**), and we also updated the Discussion in accordance with the new results.

All changes in the text are highlighted in **blue** in the manuscript.

Reviewer 1:

"The authors have responded appropriately to all my comments. In particular, adding the details of the models have clarified some of the issues that I had".

We would like to thank the reviewer once again for their previous insightful comments, which have greatly improved our manuscript.

"One minor comment is that the stochastic synaptic model resembles one studied previously: <https://www.sciencedirect.com/science/article/pii/S0925231204000578>".

The model of synaptic stochasticity studied by de la Rocha et al.¹ is similar to our model, both are stochastic versions of the Tsodyks-Markram model². We updated the Methods section to more explicitly highlight the origin of our model^{3,4} to aid comparison with other studies (lines 713-714).

Reviewer 2:

"I regret that the authors have chosen to argue against my comments rather than address them more directly. The manuscript presents an impressive amount of work, yet in the present format the results are mostly descriptive."

We thank the reviewer for acknowledging the amount of work that went into the study. We would like to point out that in addition to the descriptive aspects of the present study, we went to great lengths to disentangle and explain the effects we observed: We observed a high intrinsic variability in terms of the divergence of spontaneous activity trajectories *and explained* that it is driven by synaptic noise that is amplified by underlying chaotic network dynamics. We further provide a quantitative prediction that synaptic noise dominates over other local cellular noise sources in shaping intrinsic variability. We observed that neurons in

the circuit responded reliably amid the noise and chaos, and we explained how this response is enabled by the recurrent circuitry, at the right EI-balance.

“A direct comparison between the analyses of simulations of the large scale model, and identical analyses in simplified, classical E-I networks would have been extremely valuable, and certainly appropriate for Nature Comm. The authors argue that this comparison is done in a different study, but I would have been glad to see it here.”

We fully agree with the reviewer that a direct comparison will be very valuable, however, it is beyond the scope of this study. There are many candidate simplified E-I network models to choose from, with diverse, even contradictory dynamical properties. To move forward in this direction, as the reviewer states, we are currently working on a systematic approach to derive a simplified model from the detailed model and understand the impact of different simplifying assumptions as part of a different study for which a preprint is already available⁵.

“However an important concern that has emerged in response to the comments of Reviewer 1 is that the spiking statistics in the large scale model are actually very far from Poisson, and seem very regular (it would have been nice to have more information about the statistics beyond Fano factors). This really puts into question what kind of dynamical regime the simulations are exhibiting, and how that relates to the cortex. The authors first argue that the activity in the cortex is not actually Poisson, which is certainly true, but the obtained Fano factors seem very small in comparison to any cortical data recorded in vivo at rest (cf e.g. Churchland et al 2010 for an overview that includes V1). The authors then argue that Poisson variability is not generated by recurrent connectivity but by external inputs. Now that is a strong and certainly controversial statement that would be a major result if it could be strongly substantiated (the authors show a bit of evidence for it in Fig S14, but there they show only a transient increase of Fano Factors, quite different from Churchland et al 2010). In any case, the authors seem to admit that as they do not have variable external inputs, in the main simulations their network is not in a regime that resembles the cortex, and this calls into question the relevance of the reported observations”.

We have previously shown that our circuit model can reproduce a range of studies on neocortical spontaneous and evoked dynamics in vivo⁶, such as cell-type specific responses to whisker flicks⁷, the sequential structure of spontaneous activity⁸, or population coupling⁹, all without additional sources of external activity.

We are in agreement with the reviewer that our claim that Poisson-like variability is generated by external inputs would be stronger if we could show that our model can reproduce findings similar to Churchland et al.¹⁰ when receiving variable external inputs, and we thank the reviewer for the suggestion. Following this suggestion, we designed a new simulation experiment in which a subset of thalamic fibers drive the circuit with variable external activity (different across trials), reflecting varying inputs from different brain regions (in fact, around 80% of presynaptic inputs are estimated to be from white matter projections from other cortical areas¹¹). These simulations revealed Poisson-like Fano factors, similar to

values for rat somatosensory cortex in vivo¹² (new **Supplementary Fig. 15a3,b1-2,c2**, $t < 0$ ms). Next, we fed reliable input to another subset of thalamic fibers, reflecting reliable, stimulus-specific inputs (repeated across trials). We observed that neurons in the circuit responded more reliably after stimulus onset, and that the Fano factors decreased (new **Supplementary Fig. 15a3,b1-2,c2**, $t > 0$ ms), qualitatively reproducing Churchland et al.¹⁰. These new simulations show that our model is indeed compatible with the observed Poisson-like spike count variability in vivo, and the quenching of that variability at stimulus onset. We describe these new results and their implications for the mechanisms of quenching of variability in the discussion (lines 560-582).

Interestingly, we further observed that even when the circuit responds in a Poisson-like manner, as shown in **Supplementary Fig. 15a3,b1-2,c2** ($t < 0$ ms), repeating one specific input out of the set of variable inputs led once again to very low Fano factors (**Supplementary Fig. 15a2,c1**). What we show is thus that the difference between the two cases, Poisson-like variability or not, is that in one case the input is controlled and known, and in the other it is not. It is our knowledge and control of the input itself which dictates the variability.

These new findings, triggered by the reviewer's observations, are consistent with the growing experimental evidence suggesting that a large amount of observed cortical variability can be explained by hidden variables such as inputs from other brain areas¹³. For example, there are many reports of travelling waves across cortical regions¹⁴, and *all* of cortex is strongly modulated by movement-related activity¹⁵. Taken together, this suggests that neuronal variability is often overestimated¹⁶, and that *intrinsic* variability due to local circuitry is not Poisson-like at all, as we conclude in the present study. The discussion has been updated to provide a more detailed treatment of these points (lines 560-582).

References:

1. de la Rocha, J., Moreno, R. & Parga, N. Correlations modulate the non-monotonic response of a neuron with short-term plasticity. *Neurocomputing* **58–60**, 313–319 (2004).
2. Tsodyks, M. V. & Markram, H. The neural code between neocortical pyramidal neurons depends on neurotransmitter release probability. *Proc. Natl. Acad. Sci.* **94**, 719–723 (1997).
3. Loebel, A. *et al.* Multiquantal release underlies the distribution of synaptic efficacies in the neocortex. *Front. Comput. Neurosci.* **3**, (2009).
4. Fuhrmann, G., Segev, I., Markram, H. & Tsodyks, M. Coding of Temporal Information by Activity-Dependent Synapses. *J. Neurophysiol.* **87**, 140–148 (2002).
5. Rössert, C. *et al.* Automated point-neuron simplification of data-driven microcircuit models. *ArXiv160400087 Q-Bio* (2016).
6. Markram, H. *et al.* Reconstruction and Simulation of Neocortical Microcircuitry. *Cell* (2015).

7. Reyes-Puerta, V., Sun, J.-J., Kim, S., Kilb, W. & Luhmann, H. J. Laminar and Columnar Structure of Sensory-Evoked Multineuronal Spike Sequences in Adult Rat Barrel Cortex In Vivo. *Cereb. Cortex* bhu007 (2014). doi:10.1093/cercor/bhu007
8. Luczak, A., Barthó, P., Marguet, S. L., Buzsáki, G. & Harris, K. D. Sequential structure of neocortical spontaneous activity in vivo. *Proc. Natl. Acad. Sci.* **104**, 347–352 (2007).
9. Okun, M. *et al.* Diverse coupling of neurons to populations in sensory cortex. *Nature* **521**, 511–515 (2015).
10. Churchland, M. M. *et al.* Stimulus onset quenches neural variability: a widespread cortical phenomenon. *Nat. Neurosci.* **13**, 369–378 (2010).
11. Reimann, M. W., Horlemann, A.-L., Ramaswamy, S., Muller, E. B. & Markram, H. Morphological Diversity Strongly Constrains Synaptic Connectivity and Plasticity. *Cereb. Cortex* **27**, 4570–4585 (2017).
12. Bale, M. R. & Petersen, R. S. Transformation in the Neural Code for Whisker Deflection Direction Along the Lemniscal Pathway. *J. Neurophysiol.* **102**, 2771–2780 (2009).
13. Fairhall, A. L. Whither variability? *Nat. Neurosci.* **22**, 329 (2019).
14. Muller, L., Chavane, F., Reynolds, J. & Sejnowski, T. J. Cortical travelling waves: mechanisms and computational principles. *Nat. Rev. Neurosci.* **19**, 255–268 (2018).
15. Musall, S., Kaufman, M. T., Gluf, S. & Churchland, A. K. Movement-related activity dominates cortex during sensory-guided decision making. *bioRxiv* 308288 (2018). doi:10.1101/308288
16. Masquelier, T. Neural variability, or lack thereof. *Front. Comput. Neurosci.* **7**, (2013).

Reviewers' Comments:

Reviewer #2:

Remarks to the Author:

I thank the authors for addressing in their reply the issue of low Fano Factors in their model. From the reply, I believe we agree on the following; (i) cortical activity in vivo typically displays high Fano factors; (ii) whether this high variability is generated by recurrent connectivity (as posited by the mechanism of excitation-inhibition balance) or by external inputs is still debated; (iii) in the model used in the manuscript, high variability can be generated only using strongly variable external inputs.

Reading the whole manuscript again, I have found it difficult to reconcile the text of the manuscript with the text of the reply. Specifically:

- the results section starts by stating "we simulated in vivo-like spontaneous neuronal activity".

Although external noise is used (component (d) of the sources of noise in Fig 1 b), the Fano factors are extremely low (Fig 1f). As discussed in my previous review, and apparently agreed upon in their reply by the authors, this is very different from what is generally accepted as "in vivo-like activity". For me, this calls into question all the statements about "in vivo-like activity" and variability.

- the abstract states: "we quantified this intrinsic variability using a biophysical model of rat neocortical microcircuitry with biologically realistic noise sources. We found that stochastic neurotransmitter release is a critical component of this variability". From my previous review and the authors' reply, my understanding is that instead external inputs are a the key component of the variability, as without them variability is very low (not mentioned in the abstract). So neurotransmitter release seems to account only for weak variability that exists in absence of strong noisy inputs.

- the abstract then says "weak thalamocortical stimuli can transiently overcome the chaos, and induce reliable spike times with millisecond precision". This is absolutely not surprising if the variability is very low to start with.

- the key issues seems to be the way the authors quantify variability. This is done using measures of the divergence between membrane potential traces on different trials, starting from the same initial conditions. The authors argue that two membrane potential traces diverge very rapidly, within 10-20ms, which they interpret as a sign of high variability. They also argue that this does not necessarily imply high spiking variability. However I do not see how these two statements are compatible: a stochastic process that loses memory faster than the inter-spike interval would necessarily lead to Poisson spiking. I see several possible explanations for this contradiction: (i) the membrane potential traces never really become decorrelated, as they reach a finite steady-state correlation, in contrast to a chaotic process; (ii) the apparent decorrelation is an artefact of averaging over all cells (while Fano factors are computed for each cell).

This is not an exhaustive set of comments, I just wanted to outline to most outstanding contradictions. Altogether I cannot support this manuscript for publication.

Response to Reviewer 2's comments:

We would once more like to thank the reviewer for their reading of the revised manuscript, and for their constructive feedback. We revised the abstract and main text of the manuscript to address their new comments, as described point-by-point below.

As before, all changes in the text are highlighted in blue in the manuscript.

(1) "I thank the authors for addressing in their reply the issue of low Fano Factors in their model. From the reply, I believe we agree on the following; (i) cortical activity in vivo typically displays high Fano factors; (ii) whether this high variability is generated by recurrent connectivity (as posited by the mechanism of excitation-inhibition balance) or by external inputs is still debated; (iii) in the model used in the manuscript, high variability can be generated only using strongly variable external inputs."

We thank the reviewer for summarizing these three points. **(i)** We agree that cortical activity often displays high Fano factors. However, as we pointed out before, this is a typical but not universal property, as there are several studies showing low sub-Poisson Fano factors^{1,2}, and there is also evidence that the regularity of firing patterns drastically differs across cortical regions³. For example, neurons in the primary visual cortex of cat can have Fano Factors very similar to our model (see Figure 4C by Kara *et al.*²). **(ii)** We agree that it is still debated how high Fano factors arise. However, there are several recent studies strongly suggesting that they are due to external inputs (i.e. from other brain areas)^{4,5}. Stringer *et al.* analyzed ~10'000 neurons across the whole brain and conclude that most spike-count variability that was previously described as "noise" is in fact encoding the behavioral state⁵ (at least in the mouse). **(iii)** We agree that high variability in terms of Fano factors of spike counts does not arise internally in our model, but requires variable external inputs. This result is in line with the recent observations by Stringer *et al.*⁵.

(2) "Reading the whole manuscript again, I have found it difficult to reconcile the text of the manuscript with the text of the reply. Specifically:

- the results section starts by stating "we simulated in vivo-like spontaneous neuronal activity". Although external noise is used (component (d) of the sources of noise in Fig 1 b), the Fano factors are extremely low (Fig 1f). As discussed in my previous review, and apparently agreed upon in their reply by the authors, this is very different from what is generally accepted as "in vivo-like activity". For me, this calls into question all the statements about "in vivo-like activity" and variability."

The "external" noise source is initially kept at a negligible level (**Figs. 1-3, 5-8**). By increasing this noise source to account for missing *internal* noise sources, we predict that synaptic noise is a dominant noise source over, for example, missing ion-channel noise (**Fig. 4**). The noise sources during what we call "in vivo-like" spontaneous activity are thus purely

internal. We updated the text in the results section to make it clearer that all noise is internal (lines 106-108).

We used the term “in vivo-like” to be consistent with the original publication describing the NMC-model⁶, and to highlight that it is the state of activity at which the NMC-model can reproduce many properties of in vivo activity⁶. However, as the reviewer pointed out, we can of course only predict properties that emerge from within a microcircuit; high spike-count variability, which we predict to arise from input from outside a local microcircuit in vivo, does thus not emerge in our model. We would like to thank the reviewer for pointing this out, and we consequently removed all twelve mentions of the term “in vivo-like” to avoid confusion.

(3) - *the abstract states: “we quantified this intrinsic variability using a biophysical model of rat neocortical microcircuitry with biologically realistic noise sources. We found that stochastic neurotransmitter release is a critical component of this variability”. From my previous review and the authors’ reply, my understanding is that instead external inputs are a the key component of the variability, as without them variability is very low (not mentioned in the abstract). So neurotransmitter release seems to account only for weak variability that exists in absence of strong noisy inputs.”*

We refer to variability both in terms of the *divergence* of activity (both membrane potentials and spike trajectories), and in terms of *spike-count* variability. Stochastic neurotransmitter release accounts for strong internally generated *variability in terms of chaotic divergence* of activity trajectories from identical initial conditions. Internally generated *variability in terms of spike-counts* is lower, and we do not claim otherwise anywhere in the manuscript. We updated the abstract (lines 12-14) and discussion (lines 515-516) to clarify that activity is highly variable in terms of the divergence of trajectories. We further rewrote the last paragraph of the results section (lines 158-167) and added a paragraph to the discussion (lines 529-539) to highlight this surprising dichotomy.

(4) “- *the abstract then says “weak thalamocortical stimuli can transiently overcome the chaos, and induce reliable spike times with millisecond precision”. This is absolutely not surprising if the variability is very low to start with.”*

When focusing exclusively on *spike counts*, internally generated variability indeed appears to be low, but it is in fact highly variable in terms of the *rapid divergence* of activity trajectories. We observed rapidly diverging spontaneous activity dynamics, with population trajectories that diverge from the same completely identical initial conditions (see for example **Fig. 2a**). Yet, in response to transient thalamic inputs (notably only to a subset of neurons at the borders of L23/L4 and L5/L6), activity can propagate through the circuit with remarkable reliability and precision (see for example **Fig. 6d**). To the best of our knowledge, this is the first report that shows how a cortical network model that is both highly chaotic and stochastic, with biologically-constrained noise sources, can respond with reliable spike-timing (even for neurons that are not directly innervated by the external thalamic inputs).

(5) “- the key issues seems to be the way the authors quantify variability. This is done using measures of the divergence between membrane potential traces on different trials, starting from the same initial conditions. The authors argue that two membrane potential traces diverge very rapidly, within 10-20ms, which they interpret as a sign of high variability. They also argue that this does not necessarily imply high spiking variability. However I do not see how these two statements are compatible: a stochastic process that loses memory faster than the inter-spike interval would necessarily lead to Poisson spiking. I see several possible explanations for this contradiction: (i) the membrane potential traces never really become decorrelated, as they reach a finite steady-state correlation, in contrast to a chaotic process;”

As the reviewer summarizes, we quantified variability both by the divergence of activity trajectories (both membrane potentials and spike timing), and by spike-count variability. We agree that the dichotomy of rapid divergence and low spike-count variability is surprising and contrary to expectations from more classic simplifying network models. We now highlight this dichotomy in the results (lines 158-167) and discussion (lines 529-539).

Ultimately, we are confident that the dichotomy is not artificial, but can be adequately explained by the biological detail of the model in ways that have been previously characterized in the literature:

One of the main reasons for the different dynamics between the NMC-model and more classic network models is neural adaptation, including both spike-frequency adaptation and synaptic adaptation. The NMC-model has highly adapting neuron models and dynamic synapse models with diverse synaptic types. Spike-frequency adaptation decreases spike-count variability⁷⁻¹⁰, and synaptic adaptation can further regularize population activity¹¹. We updated the results section (lines 164-167) and discussion (lines 535-537) to point this out explicitly.

Previous theoretical and computational work has shown that cortical neurons cannot be perfectly decorrelated when biological details of cortical circuits such as distance-dependent connectivity, distinct excitatory and inhibitory synaptic timescales, and heterogeneous connectivity (all part of the NMC-model) are taken into account¹²⁻¹⁵. Yet, activity in the microcircuit is clearly chaotic, in the sense that any small perturbation will lead to diverging trajectories, even when all noise sources are turned off (see **Fig. 3**).

“(ii) the apparent decorrelation is an artefact of averaging over all cells (while Fano factors are computed for each cell).”

Both divergence and Fano factors are computed for each cell (see **Supplementary Fig. 1**), and the population firing rate diverges rapidly as well (see **Fig. 2a, Supplementary Fig. 4**).

“This is not an exhaustive set of comments, I just wanted to outline to most outstanding contradictions. Altogether I cannot support this manuscript for publication.”

We thank the reviewer for reading our manuscript once again, and raising their perceived contradictions. We revised the text accordingly to clarify our findings and conclusions.

References:

1. Hires, S. A., Gutnisky, D. A., Yu, J., O’Connor, D. H. & Svoboda, K. Low-noise encoding of active touch by layer 4 in the somatosensory cortex. *eLife* **4**, e06619 (2015).
2. Kara, P., Reinagel, P. & Reid, R. C. Low Response Variability in Simultaneously Recorded Retinal, Thalamic, and Cortical Neurons. *Neuron* **27**, 635–646 (2000).
3. Shinomoto, S. *et al.* Relating Neuronal Firing Patterns to Functional Differentiation of Cerebral Cortex. *PLOS Comput. Biol.* **5**, e1000433 (2009).
4. Musall, S., Kaufman, M. T., Gluf, S. & Churchland, A. K. Movement-related activity dominates cortex during sensory-guided decision making. *bioRxiv* 308288 (2018). doi:10.1101/308288
5. Stringer, C. *et al.* Spontaneous behaviors drive multidimensional, brainwide activity. *Science* **364**, eaav7893 (2019).
6. Markram, H. *et al.* Reconstruction and Simulation of Neocortical Microcircuitry. *Cell* (2015).
7. Fuhrmann, G., Markram, H. & Tsodyks, M. Spike Frequency Adaptation and Neocortical Rhythms. *J. Neurophysiol.* **88**, 761–770 (2002).
8. Lundstrom, B. N., Higgs, M. H., Spain, W. J. & Fairhall, A. L. Fractional differentiation by neocortical pyramidal neurons. *Nat. Neurosci.* **11**, 1335–1342 (2008).
9. Farkhooi, F., Muller, E. & Nawrot, M. P. Adaptation reduces variability of the neuronal population code. *Phys. Rev. E* **83**, 050905 (2011).
10. Farkhooi, F., Froese, A., Muller, E., Menzel, R. & Nawrot, M. P. Cellular Adaptation Facilitates Sparse and Reliable Coding in Sensory Pathways. *PLOS Comput. Biol.* **9**, e1003251 (2013).
11. Fuhrmann, G., Segev, I., Markram, H. & Tsodyks, M. Coding of Temporal Information by Activity-Dependent Synapses. *J. Neurophysiol.* **87**, 140–148 (2002).
12. Rosenbaum, R., Smith, M. A., Kohn, A., Rubin, J. E. & Doiron, B. The spatial structure of correlated neuronal variability. *Nat. Neurosci.* **20**, 107–114 (2017).
13. Stringer, C. *et al.* Inhibitory control of correlated intrinsic variability in cortical networks. *eLife* **5**, e19695 (2016).
14. Huang, C. *et al.* Circuit Models of Low-Dimensional Shared Variability in Cortical Networks. *Neuron* (2018). doi:10.1016/j.neuron.2018.11.034
15. Landau, I. D., Egger, R., Dercksen, V. J., Oberlaender, M. & Sompolinsky, H. The Impact of Structural Heterogeneity on Excitation-Inhibition Balance in Cortical Networks. *Neuron* **92**, 1106–1121 (2016).